# Fast-spiking interneuron detonation drives high-fidelity inhibition in the olfactory bulb

**Shawn D. Burton** [ID]*, **Christina M. Malyshko, Nathaniel N. Urban**

Department of Biological Sciences, Lehigh University, Bethlehem, Pennsylvania, United States of America

* shb420@lehigh.edu

**Data Availability Statement:** Individual data points are available as downloadable Supporting Information files with legends.

## Abstract

Inhibitory circuits in the mammalian olfactory bulb (OB) dynamically reformat olfactory information as it propagates from peripheral receptors to downstream cortex. To gain mechanistic insight into how specific OB interneuron types support this sensory processing, we examine unitary synaptic interactions between excitatory mitral and tufted cells (MTCs), the OB projection neurons, and a conserved population of anaxonic external plexiform layer interneurons (EPL-INs) using pair and quartet whole-cell recordings in acute mouse brain slices. Physiological, morphological, neurochemical, and synaptic analyses divide EPL-INs into distinct subtypes and reveal that parvalbumin-expressing fast-spiking EPL-INs (FSIs) perisomatically innervate MTCs with release-competent dendrites and synaptically detonate to mediate fast, short-latency recurrent and lateral inhibition. Sparse MTC synchronization supralinearly increases this high-fidelity inhibition, while sensory afferent activation combined with single-cell silencing reveals that individual FSIs account for a substantial fraction of total network-driven MTC lateral inhibition. OB output is thus powerfully shaped by detonation-driven high-fidelity perisomatic inhibition.

## Introduction

Circuit operations underlying mammalian brain function critically depend on the diverse structural and functional features of distinct inhibitory interneuron types, such as the dynamic modulation of projection neuron gain, spike-time patterning, and synchronization by fast-spiking perisomatic-innervating basket cells in neocortex and hippocampus [1–3]. Resolving the functions of interneurons in these circuits through cell type-specific recordings of unitary synaptic interactions has dramatically advanced our understanding of brain function across learning and disease [4]. In the olfactory bulb (OB), the first processing station of the main olfactory system, inhibitory interneurons likewise exhibit pronounced diversity [5,6] and are similarly central to OB circuit operations, with perturbations of synaptic inhibition significantly disrupting projection mitral and tufted cell (MTC) spike-time patterning and olfactory-guided behavior [7–9]. Compared to the extensive progress made in linking diverse interneuron types to specific functions elsewhere in the brain, however, understanding how specific inhibitory interneurons support OB circuit operations remains more limited.

**Funding:** This work was supported by National Institute on Deafness and Other Communication Disorders (https://www.nidcd.nih.gov/) grants R01DC016560 to N.N.U. and R01DC021296 to S. D.B. The funders had no role in study design, data collection and analysis, decision to publish, or preparation of the manuscript.

**Competing interests:** The authors have declared that no competing interests exist.

**Abbreviations:** ADP, afterdepolarization; AHP, afterhyperpolarization; dTC, deep tufted cell; EPL, external plexiform layer; EPL-IN, external plexiform layer interneuron; FSI, fast-spiking interneuron; GC, granule cell; IPSC, inhibitory postsynaptic current; IPSP, inhibitory postsynaptic potential; IR-DIC, infrared differential interference contrast; ISI, interspike interval; LY, Lucifer Yellow; MC, mitral cell; MTC, mitral and tufted cell; OB, olfactory bulb; OMP, olfactory marker protein; PB, phosphate buffer; PFA, paraformaldehyde; PV, parvalbumin; RSI, regular-spiking interneuron; sTC, superficial tufted cell; TC, tufted cell; VIP, vasoactive intestinal peptide.

Central olfactory processing begins with transmission of sensory information from peripheral sensory neuron terminals to MTCs within odorant receptor-specific glomeruli in the OB (Fig 1A). Diverse juxtaglomerular interneurons, including large populations of inhibitory periglomerular cells [10], modulate this transmission to collectively support sniff-frequency coupling, gain control, and intraglomerular contrast enhancement of MTC odor responses [5,11,12]. Other circuit operations critical to olfaction, such as fast timescale spike-time patterning and synchronization of MTC activity, emerge deeper in the OB, within the external plexiform layer (EPL) [13], where conceptual models focus exclusively on the contribution of inhibitory granule cells (GCs) to recurrent and lateral MTC inhibition. Despite theoretical and histological evidence of widespread MTC–GC connections [14,15], however, functional unitary MTC–GC connectivity appears exceedingly sparse, with <5% of pair recordings exhibiting MTC-to-GC excitation and zero exhibiting GC-to-MTC inhibition [16–18]. Investigation of MTC inhibition in the EPL has consequently relied on non-cell type-specific measures of total recurrent inhibition evoked by prolonged MTC activation, assumed to originate from GCs, and often performed in low $Mg^{2+}$ to augment GC excitation [5,19]. These measures have led to the consensus that MTC inhibition is slow and low-fidelity [20,21], features difficult to reconcile with functions such as the precise regulation of MTC spike timing.

Fundamental understanding of the circuit operations underlying olfaction will thus ultimately require greater cell type-specific knowledge of unitary synaptic interactions in the OB. Of paramount interest, the EPL contains a conserved and neurochemically diverse population of anaxonic interneurons (EPL-INs) that, while less abundant than GCs, can mediate unitary MTC inhibition [17,22]. Population-level manipulations further suggest that such inhibition powerfully influences MTC activity [17,22–24]. However, interpretation of these genetically targeted approaches remains constrained by disruption of endogenous expression in at least some mouse lines [23,25,26], nonselective perturbation of other neuron types [17,23,27,28], incomplete mapping of neurochemical identity across EPL-IN subtypes, and limited mechanistic insight into unitary interactions, leaving the overall function of EPL-INs still unclear.

To advance our fundamental understanding of circuit operations in the OB, we have therefore performed a systematic investigation of unitary synaptic interactions between MTCs and EPL-INs in acute slices using simultaneous whole-cell recordings in physiological $Mg^{2+}$. Our results reveal that fast-spiking EPL-INs perisomatically innervate MTCs with noncanonical architecture and mediate a substantial fraction of total MTC inhibition through fast, synchronous synaptic release. This release, supporting high-fidelity recurrent and lateral inhibition, is driven by both unusually prevalent synaptic detonation and high sensitivity to sparse MTC synchronization. Collectively, these findings challenge multiple conceptual paradigms of OB circuit operation and provide new insight into key modes of inhibitory signaling supporting olfaction.

## Results

### EPL-INs comprise 2 subtypes with distinct circuit functions

To systematically investigate unitary MTC–EPL-IN interactions independent of neurochemical identity and molecular lineage, we targeted MTCs and nearby small somata of putative EPL-INs for whole-cell pair recordings in acute OB slices prepared from wild-type mice (see Materials and methods). Despite wide neurochemical heterogeneity noted across EPL-IN subsets (e.g., [22,29]), analysis of intrinsic biophysical properties across 145 MTC–EPL-IN pairs surprisingly revealed only 2 major EPL-IN subtypes: (1) fast-spiking interneurons (FSIs) with non-adapting spike trains, high instantaneous firing rates, and frequent spike clustering; and (2) regular-spiking interneurons (RSIs) with regular, adapting, and comparatively slower firing

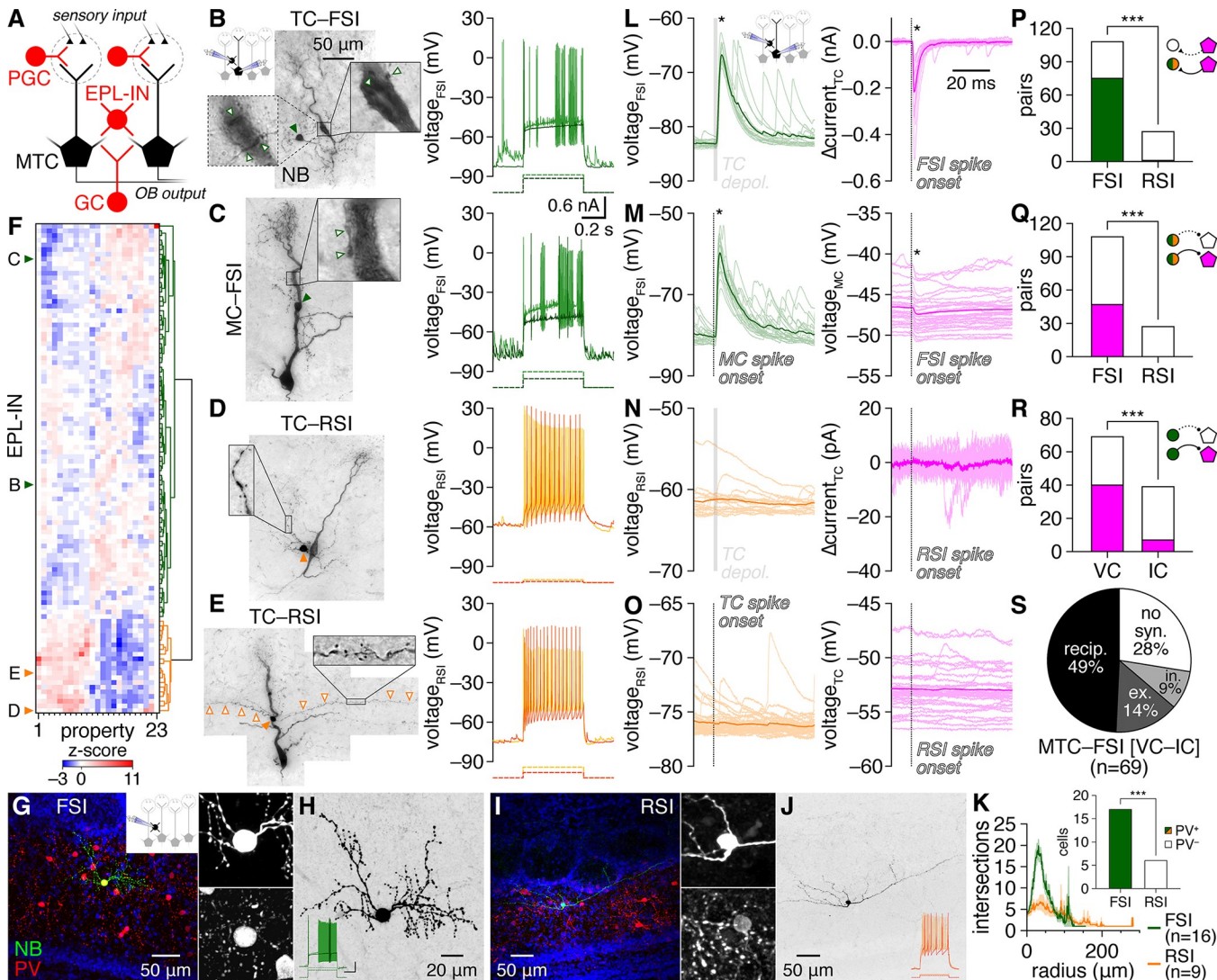

**Fig 1. EPL-INs subdivide into fast- and regular-spiking subtypes.** (**A**) Schematic of major OB components supporting feedforward, recurrent, and lateral MTC inhibition. Within glomeruli (dashed circles), peripheral sensory neuron terminals excite both MTCs and PGCs, driving intraglomerular feedforward inhibition of MTCs. Within the EPL, both EPL-INs and GC apical dendrites can form dendrodendritic synapses with MTCs to support potentially distinct modes of recurrent inhibition and lateral inhibition on interglomerular scales. Glutamatergic neurons shown in black; GABAergic neurons shown in red. See also [5] for a more comprehensive version containing other circuit components not directly studied here. (**B, C**) Example MTC–FSI pairs filled with NB (left) and interneuron fast-spiking physiology (right) in response to step current injections (dashed lines). Magnified regions here and throughout other figures shows single optical planes of example putative MTC–FSI synaptic contact (open arrowheads), except where noted. Schematic of recording configuration (inset) shown here and throughout other figures when configuration changes. (**D, E**) Example MTC–RSI pairs and regular-spiking physiology. Magnified region: long spiny RSI dendrites (open arrowheads). Filled arrowheads: interneuron somata. (**F**) Hierarchical clustering of 104 EPL-INs by intrinsic biophysical properties into FSIs (green) and RSIs (orange). (**G**) Post hoc PV staining of an example FSI. Insets: magnified region centered on soma, showing NB (upper) and PV (lower). (**H**) NB signal from **G**, inverted to show morphology. Inset: spiking physiology (scale: 20 mV/0.5 nA, 0.2 s). (**I, J**) Same as **G and H** for an RSI. (**K**) Sholl analysis. Inset: FSIs were PV$^+$, while RSIs were PV$^-$ (***$p = 1.6 \times 10^{-6}$, $\chi^2_{[1]} = 23$, $\chi^2$ test). (**L–O**) Unitary synaptic interactions from the pairs in **B–E**. Asterisks mark significant postsynaptic responses. (**P, Q**) Distribution of excitatory (**P**) (***$p = 7.3 \times 10^{-10}$, $\chi^2_{[1]} = 37.9$, $\chi^2$ test) and inhibitory (**Q**) (***$p = 2.2 \times 10^{-5}$, $\chi^2_{[1]} = 18.0$, $\chi^2$ test) unitary connections among MTC–FSI and MTC–RSI pairs. Simplified schematic here and elsewhere indicates neuron types being plotted by color (MTC: magenta; FSI: green; RSI: orange) and shape (MTC: pentagon with triangular terminal; EPL-IN: circle with circular terminal), as well as presence vs. absence of significant postsynaptic response (solid connection and coloration vs. dotted connection and no coloration). (**R**) Voltage-clamp (VC) detected unitary inhibitory connections more sensitively than current-clamp (IC) (***$p = 5.6 \times 10^{-5}$, $\chi^2_{[1]} = 16.2$, $\chi^2$ test). (**S**) Distribution of inhibitory, excitatory, reciprocal, or no unitary connectivity within the subset of MTC–FSI pairs recorded in VC and IC, respectively. Source data for panels F, K, and P–S are provided in Supporting information, S1 Data. EPL-IN, external plexiform layer interneuron; FSI, fast-spiking interneuron; GC, granule cell; MTC, mitral and tufted cell; OB, olfactory bulb; PGC, periglomerular cell; PV, parvalbumin; RSI, regular-spiking interneuron.

(Fig 1B–1E). Unbiased hierarchical clustering (Fig 1F) and principal component analysis (S1 Fig) of 23 intrinsic biophysical properties directly supported this classification, with FSIs and RSIs exhibiting stark differences across the majority of properties (S2 Fig and S1 Table).

As a caveat, such hierarchical clustering approaches can be sensitive to the specific algorithms and parameter spaces used. Reinforcing the division of EPL-INs into 2 major subtypes, however, FSIs and RSIs also exhibited clear morphological differences: FSIs extended complex and beaded dendritic arbors while RSIs extended less branched and long, spiny dendrites that occasionally entered the glomerular layer (Figs 1B–1E, 1G–1K, S3, S4, and S5 and S2 Table). Neither FSIs nor RSIs extended visible axons. Morphological properties of FSIs thus broadly matched those previously noted for several neurochemical EPL-IN subsets [17,22,29,30], while RSIs did not obviously correspond to any previously characterized EPL-INs. Consistent with these parallels, post hoc immunostaining confirmed parvalbumin (PV) expression in 100% of FSIs and 0% of RSIs (Figs 1G, 1I, 1K, and S3). Likewise, neither FSIs nor RSIs expressed tyrosine hydroxylase (TH), a marker of sparse, tonically active short-axon cells in the EPL (S4 Fig) [31–33]. How other neurochemicals map to FSIs versus RSIs remains unclear, however, with only 27% of FSIs exhibiting weakly-detectable vasoactive intestinal peptide (VIP) expression compared to 0% of RSIs (S5 Fig).

Physiological and morphological EPL-IN differences also mapped directly onto unitary synaptic connectivity evoked by single spikes or brief voltage steps (see Materials and methods): MTC activation triggered robust postsynaptic depolarization of FSIs but not RSIs, while FSI but not RSI spiking evoked inhibitory postsynaptic currents (IPSCs) in voltage-clamped MTCs and inhibitory postsynaptic potentials (IPSPs) in current-clamped MTCs (Fig 1L–1O). Across 145 pairs examined, MTCs were exclusively inhibited by FSIs (Fig 1Q) while 69% of FSIs exhibited postsynaptic excitation (Fig 1P) and only 1 of 27 RSIs exhibited significant excitation that was orders of magnitude weaker (S6 Fig). Lack of MTC–RSI connectivity could not be attributed to any distance-dependent slicing artifact, as MTC–RSI pairs exhibited modestly shorter intersomatic distances than MTC–FSI pairs (S7 Fig). Glutamate receptor antagonists NBQX and AP5 reversibly blocked unitary MTC-to-FSI excitation, while additional application of $GABA_A$ receptor ($GABA_A$R) antagonist gabazine reversibly blocked unitary FSI-to-MTC inhibition (S8 Fig), confirming that unitary MTC–FSI interactions are mediated by direct glutamatergic and GABAergic transmission. In total, physiological, morphological, neurochemical, and synaptic differences thus subdivide EPL-INs into 2 major subtypes with profoundly distinct circuit functions.

The majority of MTC–FSI pairs with any connection proved reciprocally connected (Fig 1S). Unitary connectivity could further be detected between FSIs and MTCs with apical dendrites truncated at the glomerular layer (S9 Fig), confirming that MTC–FSI connectivity is localized to infraglomerular layers. No electrical coupling was detected between MTCs and FSIs (see Materials and methods). Across the EPL, unitary connection probability and strength did not differ between FSIs and mitral cells (MCs) versus tufted cells (TCs) (S10 Fig). Likewise, MCs and TCs interacted with FSIs spanning comparable positions throughout the EPL (S10 Fig). FSIs are thus capable of influencing both streams of olfactory processing separately supported by TCs and MCs [9,34–36], and results are thus reported as they apply to the collective MTC population.

Notably, the unitary synaptic output of both MTCs and FSIs exhibited short latencies, low jitter, and high release probability (S11 Fig), signatures of high-fidelity, synchronous synaptic release that markedly differ from the low-fidelity, asynchronous release predicted from GCs [37]. Further of note, unitary FSI-to-MTC inhibition exhibited consistently shorter latencies than unitary MTC-to-FSI excitation (S11 Fig), potentially indicating distinct presynaptic $Ca^{2+}$ channel expression [38]. Unitary MTC-to-FSI excitation also exhibited higher release

probabilities on average than unitary FSI-to-MTC inhibition (S11 Fig), though prodigious spontaneous EPSP (sEPSP) rates in FSIs (S1 Table) likely masked some MTC release failures. Unitary EPSP (uEPSP) and unitary IPSC (uIPSC) amplitudes also positively correlated among connected MTC–FSI pairs (S11 Fig), suggesting coordinated scaling of synaptic strength.

In total, our results thus identify FSIs as a major EPL-IN subtype with strong potential to directly influence OB output through prevalent, reciprocal, and fast unitary synaptic interactions with MTCs. We therefore focused on MTC–FSI interactions throughout the remainder of the study.

### FSIs perisomatically innervate MTCs with noncanonical architecture

Post hoc inspection of reciprocally connected MTC–FSI pairs filled with Neurobiotin (NB) revealed FSI dendritic varicosities apposed to MTC somata, proximal apical dendrites, and axon hillocks (Fig 1B and 1C)—subcellular domains matching the description of perisomatic innervation of projection neurons elsewhere in the brain [39]. Post hoc multicolor confocal microscopy of an additional subset of reciprocally connected MTC–FSI pairs filled with Lucifer Yellow (LY) and NB, respectively, likewise resolved putative contact between FSI dendrites and MTC perisomatic domains (Fig 2A and 2B). These patterns directly complement previous ultrastructural investigations of PV$^+$ EPL-INs [40–42], suggesting that FSIs perisomatically innervate MTCs to support similar circuit operations as perisomatic-innervating interneurons elsewhere in the brain, such as PV$^+$ basket cells in neocortex and hippocampus. Two lines of evidence further suggest that FSIs preferentially innervate perisomatic over distal MTC domains.

First, connected MTC–FSI pairs exhibited significantly shorter intersomatic distances than unconnected pairs (S7 Fig), with connected distances within the range for compact FSI dendrites (radially extending approximately 100 μm; Fig 1K) to contact MTC somata. This distance-dependence would be unlikely to emerge from our data if FSIs innervated distal and proximal MTC domains equally. Further, neither uEPSP nor uIPSC amplitudes correlated with intersomatic distance (S7 Fig), suggesting that the greater connectivity observed at shorter distances did not reflect a technical inability to resolve potentially weaker distal connections. While additional pair recordings will ultimately be required to assess the distance-dependence of MTC–FSI connectivity across intersomatic distances encompassing the full ~1 mm extent of MTC lateral dendrites (see Discussion), the high rates of connectivity between nearby MTCs and FSIs, combined with the compact dendritic arbors of FSIs, provides a morphological foundation for preferential FSI innervation of perisomatic MTC domains.

Second, unitary FSI-to-MTC inhibitory events exhibited significantly faster rise times than spontaneous events in 52% of connected pairs and significantly slower rise times in 0% (Fig 2C and 2D), consistent with FSIs innervating MTCs at more electrotonically proximal domains than inhibition from other interneurons [43], such as GCs and juxtaglomerular interneurons. In comparison, MTC-to-FSI uEPSP rise times were significantly faster than sEPSPs in only 1% of connected pairs and significantly slower in only 6% (Fig 2C and 2D), suggesting that nearby MTCs comparably innervate proximal and distal domains of FSIs. While other factors may contribute to faster MTC inhibition by FSIs than other interneurons (see Discussion), the combined observation of faster events, MTC–FSI connectivity at short intersomatic distances, and apposition of FSI dendritic varicosities to postsynaptic MTC somata, proximal apical dendrites, and axon hillocks complements prior ultrastructural findings to suggest that FSIs preferentially (though not exclusively) mediate perisomatic MTC inhibition.

To gain further structural insight into perisomatic MTC inhibition, we next examined whether FSIs, which share several neurochemical and physiological properties with PV$^+$ fast-

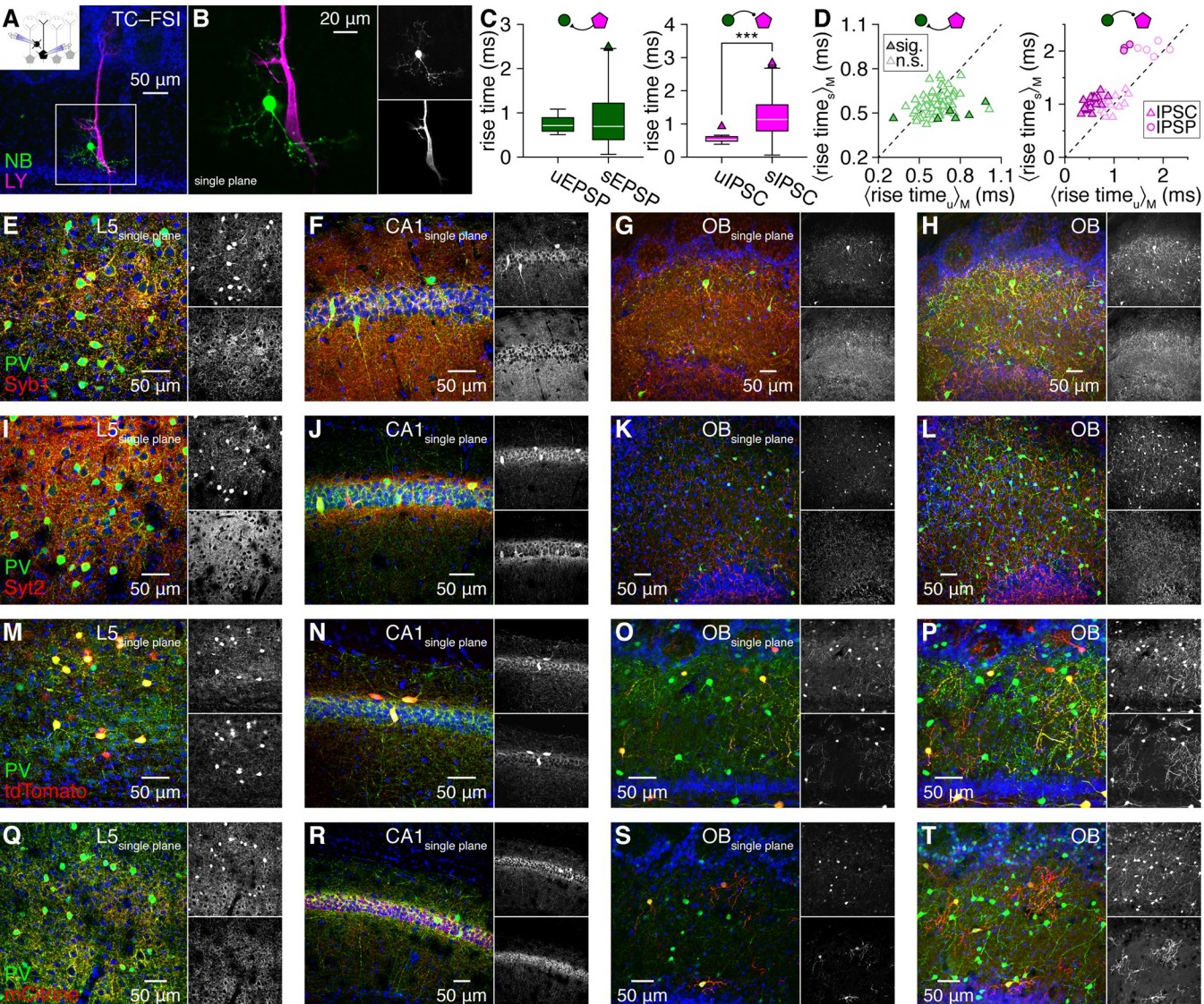

**Fig 2. FSIs mediate fast MTC inhibition through perisomatic innervation lacking basket-like structures.** (**A**) Example reciprocally coupled MTC–FSI pair filled with LY and NB. (**B**) Magnified region from **A**, showing single optical plane (**B**). (**C**) MTC-to-FSI uEPSPs ($n$ = 14 event waveforms) exhibited comparable rise times as sEPSPs ($n$ = 3,965 event waveforms) in the pair in **A and B** (left; $p$ = 0.8, r.s. = 7889511, Wilcoxon rank-sum test with Bonferroni-corrected significance level of 0.05/78), while FSI-to-MTC uIPSCs ($n$ = 23 event waveforms) exhibited faster rise times than sIPSCs ($n$ = 613 event waveforms) (right; ***$p$ = 5.3 × 10$^{-11}$, r.s. = 200918.5, Wilcoxon rank-sum test with Bonferroni-corrected significance level of 0.05/52). (**D**) Comparison of unitary and spontaneous synaptic event kinetics as in **C** (only median rise times shown for visual clarity) for 78 pairs exhibiting unitary MTC-to-FSI excitation (left) and 52 pairs exhibiting unitary FSI-to-MTC inhibition (right). (**E, F**) Single optical confocal planes of layer 5 (L5) primary visual neocortex (**E**) and hippocampal CA1 (**F**) revealing colocalization of Syb1 and PV in basket-like outlines of putative pyramidal cells. (**G, H**) Single optical confocal plane (**G**) and maximum-intensity projection (approximately 50 μm depth) (**H**) of Syb1 and PV in the OB, revealing a consistent absence of basket-like outlines of MTCs. (**I–L**) Same as **E–H** for Syt2 and PV. (**M–T**) Same as **E–H** for PV and Cre-dependent expression of either cytosolic tdTomato (**M–P**) or membrane-localized mCitrine (**Q–T**) in PV-IRES-Cre mice. Source data for panels C and D are provided in Supporting information, S2 Data. FSI, fast-spiking interneuron; LY, Lucifer Yellow; MTC, mitral and tufted cell; OB, olfactory bulb; PV, parvalbumin; sIPSC, spontaneous IPSC.

spiking basket cells throughout several regions of the brain, including neocortex and hippocampus, also innervate MTCs with canonical patterns of profuse basket-like somatic wrapping similar to basket cell innervation of pyramidal cells. Toward this end, we first immunostained for Synaptobrevin-1 (Syb1; also known as VAMP1) and Synaptotagmin-2 (Syt2), 2 proteins co-expressed with PV in neocortical and hippocampal basket cell axonal boutons [44,45].

Consistent with prior results, Syb1 and Syt2 colocalized with PV in canonical basket-like outlines of PV⁻ putative pyramidal cells in single optical confocal planes of both neocortex and hippocampus (Fig 2E, 2F, 2I, and 2J). In the same sections, we detected clear expression of both Syt2 and Syb1 in the OB, in contrast to reports suggesting negligible OB expression [46,47]. However, while Syb1 colocalized with PV throughout the EPL, particularly within large PV⁺ somata likely corresponding to a sparse subset of short-axon cells [48], no basket-like innervation of MTCs by PV/Syb1-coexpressing processes was evident in either single optical confocal planes or maximum-intensity projections (Fig 2G and 2H). Syt2, in turn, was also found throughout the EPL, as well as in prominent clusters throughout the mitral cell and internal plexiform layers (Fig 2K and 2L), potentially suggesting perisomatic MTC innervation via Chandelier-like structures [49]. However, these Syt2 structures not only failed to clearly associated with MTC axon initial segments (S12 Fig) but also further exhibited even more limited colocalization with PV than did Syb1 (Fig 2K and 2L).

While immunostaining for PV and presynaptic basket cell proteins Syt2 and Syb1 thus failed to reveal canonical basket-like innervation patterns, FSIs may innervate MTC somata with alternative presynaptic proteins and fine dendritic processes that stain poorly for PV. Indeed, comparison of intracellular NB and post hoc PV immunostaining revealed incomplete PV staining of fine FSI dendrites (Figs 1G and S3). We therefore additionally explored whether genetic labeling of PV⁺ interneurons using crosses of PV-IRES-Cre mice and Cre-dependent cytosolic tdTomato or membrane-localized mCitrine reporter mice may reveal basket-like innervation of MTCs. In neocortex and hippocampus, cytosolic tdTomato brightly labeled PV⁺ somata and, in hippocampus, also outlined PV⁻ putative pyramidal cell somata (Fig 2M and 2N), while membrane-localized mCitrine could not clearly be visualized in PV⁺ somata but extensively colocalized with PV in basket-like innervation structures (Fig 2Q and 2R). Surprisingly, PV-IRES-Cre mice crossed to either reporter only sparsely labeled PV⁺ EPL-INs in the OB, in addition to PV⁺ periglomerular cells and deep short-axon cells (Fig 2O, 2P, 2S, and 2T)—a level of labeling inefficiency similar to some other brain regions [50]. Nevertheless, both reporters revealed fine dendritic processes of the sparsely labeled EPL-INs, but still failed to uncover signs of even partial basket-like MTC innervation.

Collectively, our results thus reveal kinetically fast perisomatic inhibition of MTCs by FSIs as a prominent feature of the mouse OB, uncovering new potential for functional parallels between OB and cortical circuits. This common functional feature surprisingly emerged through noncanonical architecture void of basket-like somatic innervation, however, highlighting key structural differences between basket cells and anaxonic FSIs that may augment established computational roles of perisomatic inhibition with the potential for independent subcellular processing within release-competent FSI dendrites (see Discussion).

### FSI detonation mediates high-fidelity inhibition

Recurrent MTC inhibition powerfully regulates odorant discrimination and olfactory perception [51] and is widely accepted to be slow and asynchronous. This low-fidelity signaling has been traced to both intrinsic and synaptic properties of GCs [5], including asynchronous GC output onto distal MTC lateral dendrites [37] as well as weak MTC excitation of GCs, with GC spiking requiring temporal summation across prolonged MTC spike trains or the synchronous spiking of 9 to 30 MTCs [18,52,53]. In contrast to such weak GC excitation, unitary MTC-to-FSI excitation triggered FSI spiking with short latency (2.20 ± 0.08 ms) and low jitter (0.36 ± 0.06 ms, $n = 12$) in 23% of connected pairs (Fig 3A and 3D; spike probability: 0.43 ± 0.08 [$n = 19$]). Identical activity was further observed spontaneously in cell-attached recordings preceding whole-cell access, with MTC spiking typical of glomerulus-wide long-

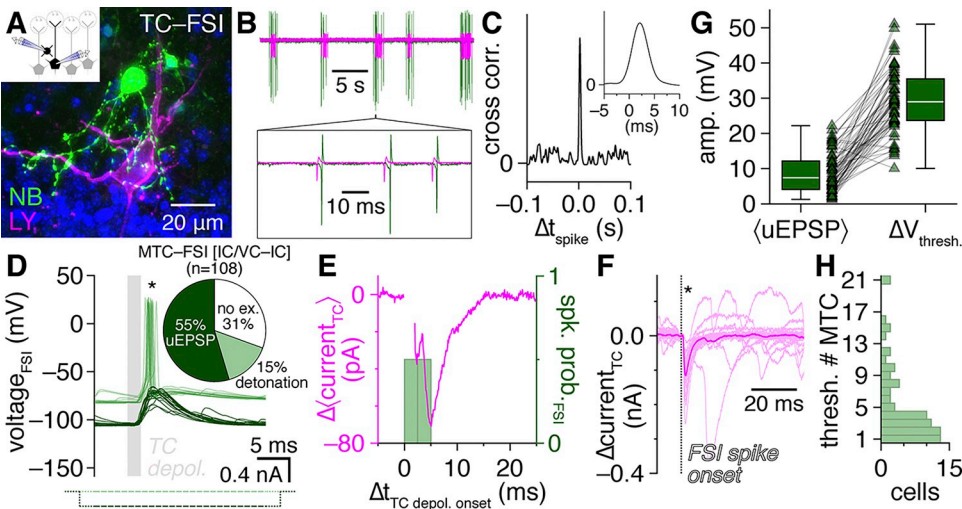

**Fig 3. FSI detonation mediates high-fidelity recurrent MTC inhibition.** (**A–C**) Cell-attached recording (**B**) preceding whole-cell access of an example MTC–FSI pair (**A**) showing FSI spikes reliably following spontaneous MTC spikes by approximately 2 ms (**B, C** insets). (**D**) Unitary MTC release triggered FSI detonation (light green) in the pair in **A**. Negative current injection (dark green) on interleaved trials blocked detonation, revealing an underlying uEPSP. Inset: proportion of MTC–FSI pair recordings showing FSI detonation, uEPSPs, or no excitation. (**E**) Subtraction of mean MTC currents across FSI detonation vs. uEPSP trials from **D** isolates IPSCs time-locked to FSI detonation. Voltage-step currents blanked for visual clarity. (**F**) Unitary FSI-to-MTC inhibition from the pair in **A**. Asterisks in **D** and **F** mark significant postsynaptic responses. (**G**) Comparison of mean MTC-to-FSI uEPSP amplitudes with $\Delta V_{thresh.}$ across 72 FSIs with detonation probability <1. (**H**) Estimated number of synchronously-spiking MTCs needed to activate each FSI (ratio of values in **G** for FSIs with detonation probability <0.5; value set to 1 for FSIs with detonation probability ≥0.5). Source data for panel D inset and panels G and H are provided in Supporting information, S3 Data. FSI, fast-spiking interneuron; IPSC, inhibitory postsynaptic current; MTC, mitral and tufted cell.

lasting depolarizations [54] reliably followed by short-latency FSI spiking (Fig 3B and 3C), thus excluding the possibility that intracellular dialysis may have artificially elevated FSI excitability.

Such rapid unitary postsynaptic activation, or detonation [55], combined with the synchronous release properties of FSIs suggests that unitary MTC release under physiological conditions may trigger high-fidelity recurrent inhibition. To investigate this, we hyperpolarized FSIs in reciprocally connected MTC–FSI pairs through negative current injection on interleaved trials, blocking detonation and revealing massive underlying uEPSPs (Fig 3D), and then examined the difference in mean MTC currents across detonation versus uEPSP trials. Of note, each MTC can interact via reciprocal dendrodendritic synapses with up to $10^4$ GCs [56,57] as well as a large population of periglomerular cells [10], suggesting that such an attempt to resolve the recurrent inhibition mediated by any one interneuron through somatic recording may prove exceedingly difficult unless such inhibition is particularly strong and reliable.

Strikingly, subtracting the mean MTC current recorded during FSI uEPSP trials from the mean MTC current recorded during detonation trials indeed isolated clear recurrent IPSCs strongly timelocked to FSI detonation latencies (Fig 3E). Subtraction-isolated MTC inhibition was further comparable to the unitary FSI-to-MTC inhibition measured separately using precisely timed FSI current injection (Fig 3F), confirming that FSI detonation alone could fully account for the difference in mean MTC currents across detonation versus uEPSP trials. Equivalent results were further observed when comparing spontaneously alternating detonation versus uEPSP trials for FSIs with detonation probability <1 (S13 Fig). FSI detonation is thus widespread throughout the OB and mediates strong, high-fidelity recurrent MTC inhibition.

While FSI detonation was surprisingly prevalent, many FSIs still responded to unitary MTC release with subthreshold excitation (Fig 3D). To comprehensively understand how the total population of FSIs contributes to MTC inhibition, we therefore divided $\Delta V_{thresh.}$ (the depolarization required to reach spike threshold; see Materials and methods) by the mean uEPSP amplitude for each FSI (Fig 3G) to estimate how many synchronously spiking MTCs would be needed to activate the FSI (assuming identical unitary strength and passive EPSP summation) (Fig 3H). Mean uEPSP amplitudes exceeded $\Delta V_{thresh.}$ in some FSIs (Fig 3G), consistent with a subset of FSIs with detonation probability <1, as well as potential differences in FSI excitation by synaptic input versus somatic current injection. Strikingly, the majority of FSIs were estimated to activate following synchronous unitary release from only 1 to 4 MTCs (Fig 3H), suggesting that FSIs prominently contribute to OB inhibition following even sparse synchronization of MTCs.

Similar to recurrent inhibition, lateral inhibition between heterotypic MTCs (i.e., connected to distinct glomeruli) is believed to support olfactory perception through key circuit operations, such as the binding of distinct percepts through the synchronization of MTC spiking [58]. While theoretical studies specifically point to fast perisomatic lateral inhibition as a powerful synchronizing force in the OB [59,60] as in other circuits [1,2], the few studies directly recording heterotypic MTC pairs have consistently reported lateral inhibition to be slow and asynchronous [20,21,61,62]. As with measures of recurrent inhibition, however, these studies evoked lateral inhibition with prolonged MTC activation, often in low $Mg^{2+}$.

Strong, high-fidelilty FSI inhibition of MTCs combined with widespread FSI detonation in physiological $Mg^{2+}$ suggests that single MTC spikes should instead evoke detectable fast lateral inhibition in nearby MTCs. Consistent with this hypothesis, re-examination of heterotypic MTC pair recordings from a recent data set [63] indeed revealed that single MTC spikes reliably evoked single, short-latency IPSPs (<10 ms) in a large proportion of nearby MTCs, including both uni- and bidirectional connections (Fig 4A–4E).

Such fast lateral inhibition is inconsistent with the weak excitation and projected asynchronous and NMDAR-dependent output of GCs [51,64]. GCs strongly outnumber EPL-INs, however, and are widely accepted to be capable of subthreshold GABA release following even weak excitation [5], and thus may concievably still drive fast lateral inhibition. Therefore, to provide a complementary test of whether FSIs versus GCs mediate fast lateral inhibition, we capitalized on the established presence versus absence of philanthotoxin-sensitive GluA2-lacking AMPARs in $PV^+$ EPL-INs versus mature GCs, respectively [16,17,65–70]. Philanthotoxin-7,4 (PhTx-74) indeed markedly reduced MTC-to-FSI uEPSP amplitudes, abolished FSI detonation (Fig 4F–4H), and disrupted fast lateral inhibition in MTC pairs (Fig 4I–4K). Our pharmacological results thus provide complementary evidence that FSI detonation plays a central role in mediating fast lateral inhibition. Further supporting this finding, we detected fast lateral inhibition in both TC pairs and MC pairs, arguing against possible contributions to fast lateral inhibition by superficial and deep short-axon cells, which selectively target TCs and not MCs [71–74].

The degree to which FSI-mediated fast lateral inhibition shapes OB activity, and in particular synchronization of MTC spiking, will critically depend on its prevalence relative to established forms of slow, presumably GC-mediated lateral inhibition following prolonged MTC activation. While our results so far suggest that fast lateral inhibition may be as prevalent as the approximately 10% of nearby MC pairs exhibiting slow lateral inhhibition [61], these data were collected under distinct conditions. We therefore next tested for the occurrence of fast versus slow lateral inhibition within heterotypic MTC pairs using short (2 ms) versus long (100 ms) presynaptic voltage steps to 0 mV to trigger single spike- versus spike train-equivalent MTC output, respectively. As slow lateral inhibition is believed to be mediated by at least

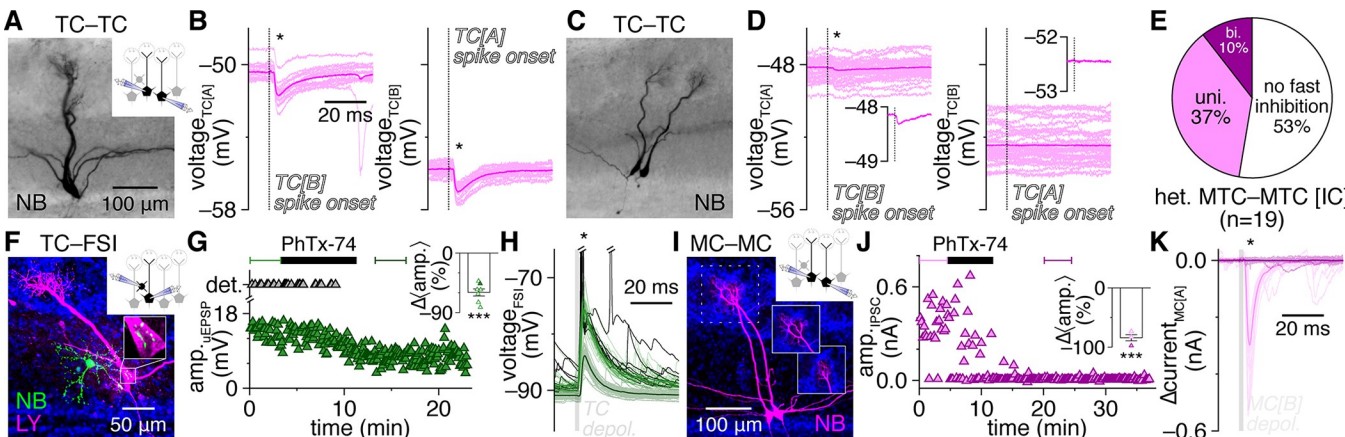

**Fig 4. FSIs mediate fast lateral inhibition between nearby heterotypic MTCs.** (**A–D**) Lateral synaptic interactions evoked by single spikes within example heterotypic MTC pairs (**A**, **C**) from [63], revealing bidirectional (**B**) and unidirectional (**D**) fast lateral inhibition (asterisks). Insets: mean postsynaptic voltages. Single widefield images shown. (**E**) Distribution of bidirectional, unidirectional, or no fast lateral inhibitory connectivity among heterotypic MTCs recorded in current-clamp (IC). (**F, G**) Example MTC–FSI pair (**F**) showing detonation occurrence and MTC-to-FSI uEPSP amplitudes before and after bath application of PhTx-74 (10 μM) (**G**). Inset: PhTx-74 significantly reduced uEPSP amplitudes (***$p = 3.4 \times 10^{-5}$, $t_6 = 11.0$, $t$ test) across 7 MTC–FSI pairs. Filled symbol corresponds to example pair. (**H**) Postsynaptic FSI voltages from the pair in **F**. Light green and black traces correspond to bracketed trials (uEPSP and detonation, respectively) before PhTx-74 application in **G**; dark green traces correspond to bracketed trials after PhTx-74 application. Asterisk marks significant unitary postsynaptic response. (**I, J**) Example MTC pair (I) (recorded in voltage-clamp to enhance the detection of inhibition), showing short-latency (<10 ms) lateral IPSC amplitudes evoked by 2-ms presynaptic depolarization before and after PhTx-74 application. Inset in **I**: sub-projection of dashed box in **I** across different depths, showing apical dendrites innervating distinct glomeruli. Inset in **J**: PhTx-74 significantly reduced fast lateral IPSC amplitudes (***$p = 4.6 \times 10^{-4}$, $t_3 = 16.8$, $t$ test) across 4 connections. Filled symbol corresponds to example pair. (**K**) Postsynaptic MTC currents from the pair in **I**. Dark and light traces correspond to the bracketed trials before and after PhTx-74 application in **J**, respectively. Asterisk marks significant fast lateral inhibition measured during pre-PhTx-74 trials. Source data for panels E, G, and J are provided in Supporting information, S4 Data. FSI, fast-spiking interneuron; IPSC, inhibitory postsynaptic current; MTC, mitral and tufted cell.

partially independent TC- and MC-innervating GC subpopulations [35] and has not been reported within mixed TC–MC pairs, we specifically targeted TC pairs and MC pairs, and not TC–MC pairs, for comparison of fast versus slow lateral inhibition prevalence. We defined fast lateral inhibition as a significant increase in IPSC probability within 10 ms following presynaptic MTC activation, while slow lateral inhibition was defined as a significant increase in IPSC probability in at least three 10 ms-bins within 250 ms following presynaptic MTC activation, reflecting an asynchronous barrage of inhibition. This approach resolved clear instances of both fast and slow lateral inhibition (Fig 5A–5C and 5D–5F, respectively), and across the population of pairs tested, fast lateral inhibition indeed proved equally as prevalent as slow lateral inhibition (Fig 5G).

Within individual MTC pairs, fast and slow lateral inhibition appeared to occur independently (though not mutually exclusively). Specifically, MTC pairs exhibiting slow lateral inhibition in response to long voltage steps typically failed to exhibit fast lateral inhibition in response to short voltage steps (Fig 5D–5F), while MTC pairs exhibiting fast lateral inhibition in response to short voltage steps typically continued to exhibit only fast lateral inhibition in response to long voltage steps (Fig 5A–5C). Our results are thus consistent with the 2 lateral inhibitory signaling modes emerging through distinct circuit mechanisms: FSI-mediated fast lateral inhibition, and presumably GC-mediated slow lateral inhibition. One MTC pair exhibited both fast and slow lateral inhibition (Fig 5H), a prevalence in line with independent occurrence of the 2 signaling modes (i.e., equal to the product of fast and slow lateral inhibitory prevalences) and suggesting that FSIs and GCs do not innervate distinct MTC subpopulations. Also of note, in a subset of connections (Fig 5G and 5H), fast lateral inhibition was only detected at the beginning of long presynaptic voltage steps (cf. Fig 5B and 5C, right); additional

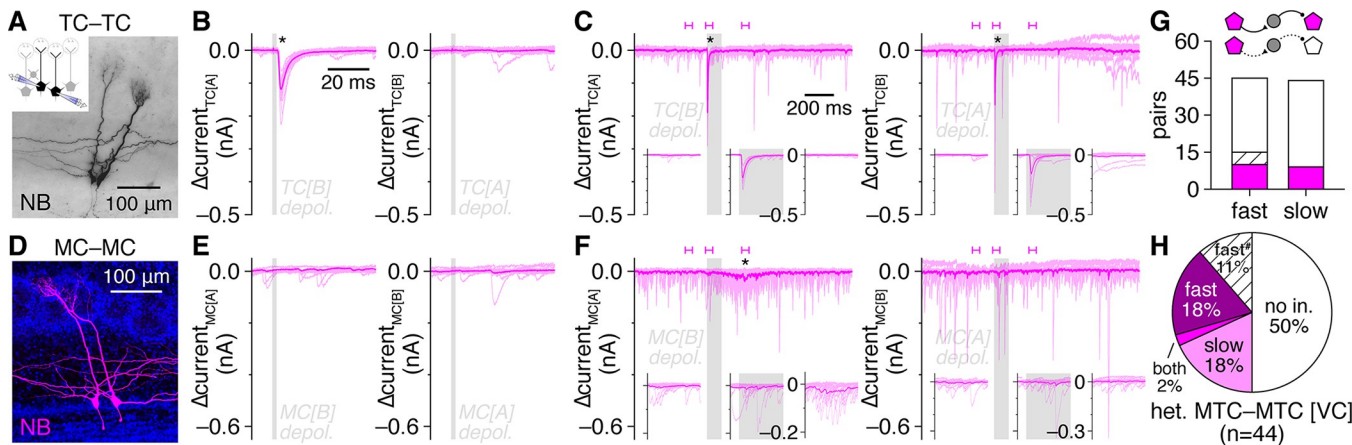

**Fig 5. Fast lateral inhibition is a prevalent signaling mode in the OB.** (**A–C**) Example MTC pair (**A**) exhibiting unidirectional fast lateral inhibition in response to 2-ms presynaptic depolarization (**B**) and bidirectional fast lateral inhibition in response to 100-ms presynaptic depolarization (**C**). Insets: enlargement of bracketed times in **C**. (**D–F**) Same as **A–C** for an example MTC pair (**D**) exhibiting unidirectional slow lateral inhibition (**F**) but no fast lateral inhibition (**E**). Asterisks mark significant fast (**B, C**) and slow (**F**) lateral inhibition. (**G**) MTC pairs exhibiting fast lateral inhibition were equally as prevalent as pairs exhibiting slow lateral inhibition ($p = 0.84$, $\chi^2_{[1]} = 4.1 \times 10^{-2}$, $\chi^2$ test). (**H**) Distribution of fast, slow, both, or no lateral inhibition within heterotypic MTC pairs recorded in voltage-clamp (VC). Pairs exhibiting fast lateral inhibition only at the start of 100-ms presynaptic depolarization (fast#) shown in hatched bars in **G** and **H**. Source data for panels G and H are provided in Supporting information, S5 Data. MTC, mitral and tufted cell; OB, olfactory bulb.

experiments are needed to determine whether such signaling reflects FSI activation by rapidly summating excitatory input. In total, our results thus reveal FSI-mediated fast lateral inhibition to be highly prevalent throughout the OB in both absolute terms and relative to presumably GC-mediated slow lateral inhibition.

Post hoc morphological inspection of the above MTC pairs revealed that both fast and slow lateral inhibition occurred independent of whether MTCs connected to directly neighboring glomeruli or glomeruli separated by 1 or ≥2 other glomeruli (S14 Fig). Moreover, intersomatic distance did not significantly differ between MTC pairs exhibiting fast versus slow lateral inhibition. Examination of each mode of inhibition in isolation, however, revealed that fast lateral inhibition was preferentially detected between nearby MTCs, while no difference in intersomatic distance was found between pairs with versus without slow lateral inhibition (S14 Fig). This difference arose largely due to a high prevalence of fast, but not slow, lateral inhibition specifically between directly neighboring MTCs (c.f. Figs 4A, 4C, 4I, and 5A). Indeed, the median intersomatic distance between MTCs exhibiting fast lateral inhibition was zero (S14 Fig). Inspection of an additional cohort of MTC pairs tested for fast lateral inhibition (see below) directly reinforced this pattern, with the probability of detecting fast, but not slow, lateral inhibition exhibiting a pronounced peak (45% of pairs) at zero intersomatic distance (S14 Fig). These results thus further reinforce the conclusion that distinct EPL circuits support fast versus slow lateral inhibition between heterotypic MTCs, with the high probability for fast lateral inhibition to link directly neighboring MTCs (in addition to other nearby MTCs at lower probability) in good agreement with the spatial dimensions of perisomatic MTC innervation by compact FSI dendritic arbors.

Odorants evoke prominent gamma-frequency oscillations throughout the OB [13], reflecting synchronization of MTC spiking [58,63,75]. The fast lateral inhibition identified in our current results represents a previously unrecognized signaling mode that may support such MTC synchronization. In turn, our estimates of MTC synchronization required to activate FSIs (Fig 3H) further suggests that synchronization of only a few MTCs should activate the

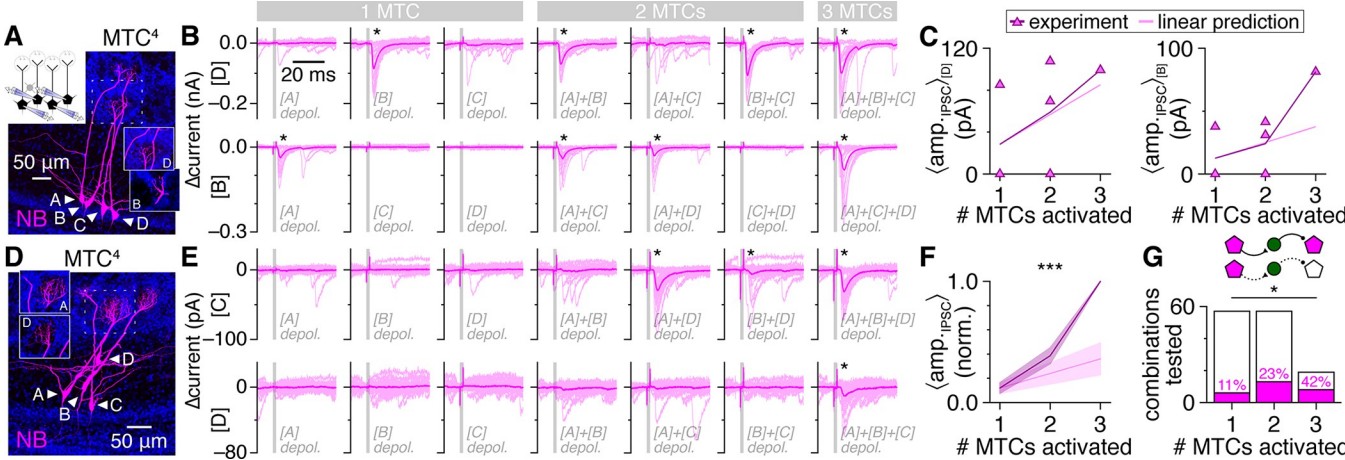

**Fig 6. Sparse MTC synchrony supralinearly enhances fast lateral inhibition strength and prevalence.** (**A, B**) Example MTC quartet (**A**; inset: sub-projection of dashed box across different depths, showing MTCs [B] and [D] innervating distinct glomeruli), showing postsynaptic currents (**B**) in MTCs [D] (upper) and [B] (lower) following singular or synchronous activation of all other quartet MTCs. Asterisks mark significant fast lateral inhibition. (**C**) Lateral IPSC amplitudes in MTCs [D] (left) and [B] (right) from the quartet in **A** as a function of the number of synchronously activated presynaptic MTCs. Lines: mean across all combinations of each presynaptic population size. (**D, E**) Same as **A and B** for a second MTC quartet. Presynaptic voltage-step artifacts are visible in postsynaptic currents of some MTCs but terminate prior to IPSC onset. (**F**) Lateral IPSC amplitudes grew supralinearly with synchronous activation of additional presynaptic MTCs (two-way ANOVA, experiment/linear prediction × # MTCs activated; ***$p = 9.3 \times 10^{-5}$, $F_{1,42} = 18.7$, experiment vs. linear prediction; ***$p = 1.0 \times 10^{-8}$, $F_{2,42} = 29.4$, # MTCs activated; ***$p = 2.4 \times 10^{-4}$, $F_{2,42} = 10.2$, interaction). Values normalized within each MTC by response to synchronous activation of 3 presynaptic MTCs and averaged across 8 MTCs exhibiting fast lateral inhibition (of 19 MTCs with no homotypic partners possible from 7 quartets; other MTCs were homotypic to ≥1 MTC of the quartet and excluded from analysis). (**G**) Fast lateral inhibition was detected across a higher proportion of combinations as more presynaptic MTCs were synchronously activated (*$p = 1.0 \times 10^{-2}$, $\chi^2_{[2]} = 9.2$, $\chi^2$ test). Source data for panels C, F, and G are provided in Supporting information, S6 Data. IPSC, inhibitory postsynaptic current; MTC, mitral and tufted cell.

majority of postsynaptic FSIs and, consequently, trigger fast lateral inhibition onto an increasing fraction of nearby MTCs to potentially spread network synchronization.

To directly test the sensitivity of FSI signaling to such MTC synchronization, we therefore next recorded MTC quartets and examined the dependence of fast lateral inhibition on the number of synchronously activated MTCs. To exclude the potential contribution of intraglomerular forms of lateral inhibition to the recorded currents [76], analysis was confined to postsynaptic responses in MTCs heterotypic to all other MTCs of the quartet (as in above MTC pair recordings). We likewise focused recordings on quartets of MCs or TCs to further parallel our MTC pair recordings above.

In some MTCs, fast lateral inhibition occurred upon activation of one presynaptic MTC regardless of whether other MTCs of the quartet were coactivated (e.g., Fig 6B, upper), a pattern consistent with FSI detonation and no additional FSI activation with MTC synchronization. In such instances, quantification of the mean lateral IPSC amplitude as a function of the number of synchronously activated MTCs closely followed linear predictions (i.e., multiplying the mean singular response by the number of MTCs coactivated) (Fig 6C, left). Other MTCs, however, exhibited supralinear increases in fast lateral inhibition, reflecting either greater-than-predicted increases in IPSC amplitude (e.g., Fig 6B, lower and 6C right) or the occurrence of fast lateral inhibition only following coactivation of 2 or 3 MTCs (e.g., Fig 6E). Across all possible combinations, both the strength and prevalence of fast lateral inhibition increased supralinearly (Fig 6F and 6G), consistent with activation of most FSIs by synchronization of just a few MTCs. Our results thus point toward a positive feedback loop between MTC synchronization and fast lateral inhibition as a potential circuit operation supported by FSIs.

Understanding the net contribution of FSIs to sensory processing in the OB will further require assessing the strength of FSI-to-MTC inhibition relative to total MTC inhibition

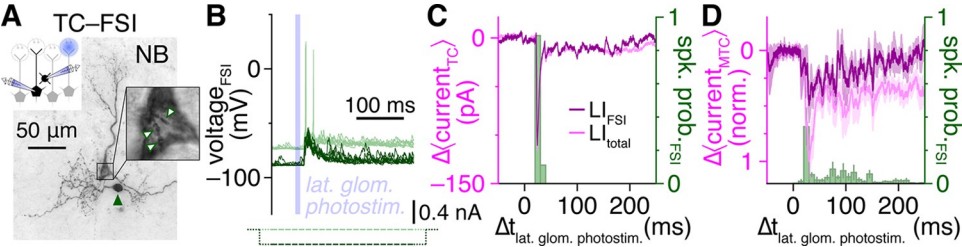

**Fig 7. Single FSIs drive a large fraction of total network-driven lateral inhibition.** (**A, B**) Example MTC–FSI pair (**A**) with suprathreshold FSI response (light green) to optogenetic photostimulation of a single lateral glomerulus (**B**). Negative current injection (dark green) on interleaved trials blocked suprathreshold FSI responses. (**C**) Subtraction of mean MTC currents across FSI suprathreshold vs. EPSP trials from **B** isolated lateral inhibitory currents timelocked to FSI spiking (LI_FSI; dark magenta). Mean MTC currents across FSI suprathreshold trials, reflecting total lateral inhibition under baseline conditions (LI_total; light magenta) shown for comparison. (**D**) Subtraction-isolated lateral inhibitory currents as in **C**, normalized within each MTC by peak LI_total current, and averaged across MTCs of 7 MTC–FSI pairs, including 2 MTCs with distal apical dendrites truncated at the glomerular layer (S9 Fig) ensuring absence of photostimulation-evoked excitatory currents. FSI, fast-spiking interneuron; MTC, mitral and tufted cell.

following sensory activation of the OB circuit, particularly given that each MTC receives inhibitory input from an estimated $10^4$ GCs [56,57] in addition to other interneuron types [5]. In our final experiment, we therefore combined MTC–FSI pair recordings with activation of glomerular sensory input to assess the contribution of single FSIs to total network-driven MTC lateral inhibition. Specifically, in acute slices prepared from olfactory marker protein (OMP)-channelrhodopsin-2 (ChR2) mice, which express ChR2 in all mature olfactory sensory neurons, including their terminals within OB glomeruli [77], we recorded MTC inhibition evoked by brief photostimulation of a single nearby lateral glomerulus while, on interleaved trials, hyperpolarizing a synaptically coupled FSI to block its feedforward excitation-evoked spiking (Fig 7A and 7B). Subtraction of mean MTC currents across FSI spiking versus EPSP trials consistently revealed robust inhibitory currents time-locked to FSI spiking (Fig 7C), regardless of whether the FSI exhibited phasic or sustained activity. Across 7 MTC–FSI pairs with suprathreshold FSI responses, activation of a single FSI remarkably accounted for 47.7 ± 12.5% of the total inhibitory charge within 250 ms of photostimulation—a typical sniff duration [78] (Fig 7D). Thus, at least under conditions physiologically mimicking single glomerulus activation of the OB with low-concentration odorants [79], high-fidelity FSI signaling accounts for a substantial fraction of total network-driven MTC lateral inhibition, suggesting that FSIs likely also play a key role in sculpting MTC tuning [80,81].

## Discussion

To advance a foundational understanding of circuit operations in the OB supporting olfactory perception, we have systematically examined unitary synaptic interactions between MTCs and EPL-INs, a highly conserved population of anaxonic OB interneurons. Challenging the consensus that MTC inhibition is slow, low-fidelity, and primarily shaped by distributed changes in activity across large populations of GCs and juxtaglomerular interneurons innervating distal MTC domains, we found that $PV^+$ fast-spiking EPL-INs perisomatically inhibit MTCs with release-competent dendrites and, through synaptic detonation and supralinear recruitment by sparse MTC synchronization, singularly mediate a substantial fraction of total MTC inhibition via strong, high-fidelity recurrent and lateral inhibition (Fig 8). These core results thus stand to reconfigure our fundamental understanding of how the OB transforms sensory input to encode olfactory information.

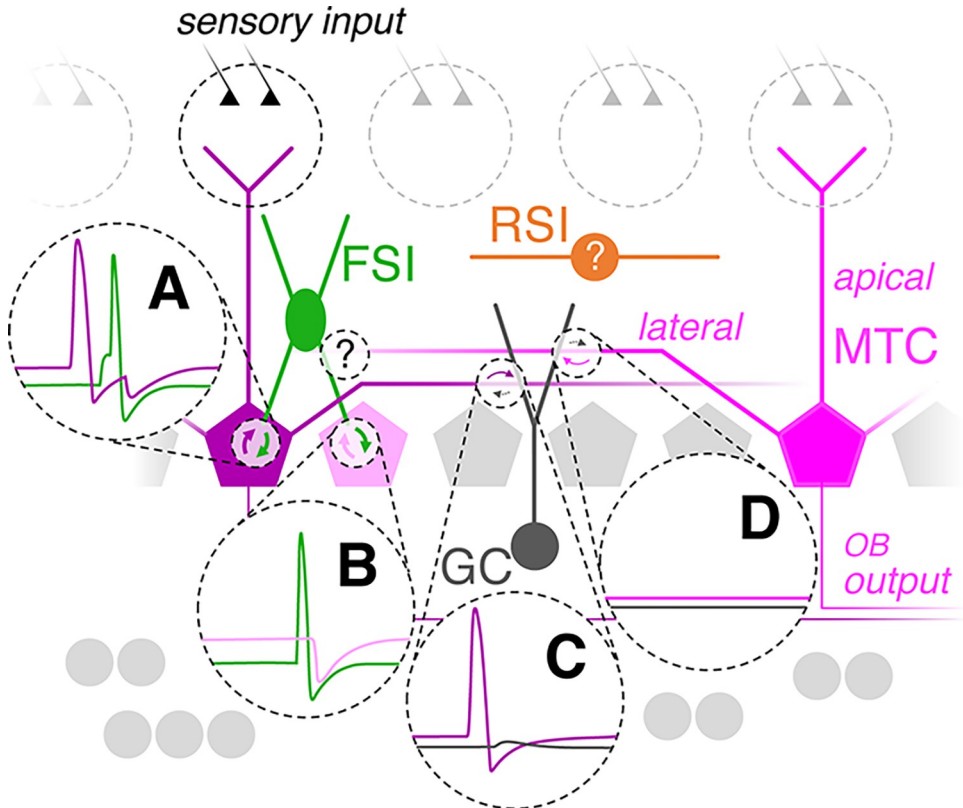

**Fig 8. FSI detonation drives high-fidelity perisomatic recurrent and lateral MTC inhibition.** Schematic of cellular voltages and synaptic events (dashed circles) underlying prominent contributions of FSIs to OB sensory processing. (**A**) MTC spiking following odor-evoked sensory input triggers strong unitary synaptic excitation and detonation of reciprocally connected FSIs, leading to fast perisomatic recurrent inhibition. (**B**) Propagation of FSI detonation drives fast perisomatic lateral inhibition of nearby MTCs via dendrosomatic synapses. (**C**) In contrast, lateral dendritic propagation of odor-evoked MTC spiking drives weak subthreshold excitation of GCs at reciprocal dendrodendritic synapses and no detectable unitary recurrent inhibition. (**D**) Weak excitation fails to propagate throughout the GC dendritic arbor, and lateral inhibition of other reciprocally connected MTCs remains gated. How distal MTCs synaptically interact with FSIs, as well as how RSIs contribute to OB circuit operation, remain open questions. FSI, fast-spiking interneuron; GC, granule cell; MTC, mitral and tufted cell; OB, olfactory bulb; RSI, regular-spiking interneuron.

## EPL-IN diversity

Classic Golgi staining has established that EPL-INs are highly conserved across mammals, including at least carnivores, omnivores, and insectivores [82–85]. Lacking functional comparison, however, these earlier studies also highlighted modest morphological differences among EPL-IN subsets of so-called "Van Gehuchten," "satellite," and "horizontal" cells. Immunohistochemistry has reinforced this apparent diversity, identifying expression of a wide variety of neurochemicals, with coexpression systematically mapped for only certain subsets (e.g., [22,29]). Collectively, these results have yielded a fairly nuanced view of EPL-INs as boutique neurons omitted from nearly all conceptual OB models.

Complementing these earlier histological approaches, we performed the first extensive physiological, morphological, neurochemical, and synaptic investigation of EPL-INs, paralleling foundational investigation of neocortical and hippocampal interneuronal diversity [86,87]. We surprisingly observed only 2 major subtypes, FSIs and RSIs, suggesting that EPL-INs are more unified in function than previously recognized. As a caveat, more subtle heterogeneity is

certain to exist. Further investigation comprehensively mapping neurochemicals across all EPL-INs together with interneuron pair recordings and more extensive post hoc staining is thus poised to uncover further contributions of EPL-INs to circuit operations in the OB, in addition to potentially overcoming limitations in genetically targeting EPL-INs (e.g., Fig 2) complicating interpretation of population-level EPL-IN manipulations.

Such investigation is likely also to establish whether the greater prevalence of FSIs than RSIs in our recordings represents a true difference in cell densities or a manifestation of recording biases. In particular, to avoid superficial granule cells or deep periglomerular cells, our recordings did not sample uniformly across the full EPL (S2 Table). Moreover, we routinely avoided targeting the largest or smallest somata for recording, as the former were likely to include at least some of the sparse EPL short-axon cells [32,33,48,88,89] while the latter proved difficult to differentiate from resealed blebs of severed MTC dendrites. Finally, for this data set we abandoned recordings from a small subset of interneurons with unstable resting membrane potential. This precaution enabled us to avoid EPL-INs damaged during slice preparation as well as the sparse subset of tonically active TH[+] short-axon cells (S4 Fig) [31–33], but may have also excluded some tonically active EPL-INs.

## Perisomatic innervation and implications of noncanonical architecture

Previous ultrastructural investigation has identified the conserved existence of perisomatic MTC innervation by certain EPL-IN subsets, including most prominently PV[+] EPL-INs [40–42]. Lacking functional confirmation, however, such targeting has not been integrated into current conceptual OB models. Here, we measured synaptic rise times and consistently found unitary FSI-to-MTC inhibition to be faster than spontaneous inhibition from other sources, which, together with post hoc structural analyses of synaptically coupled pairs, compellingly argues that FSIs perisomatically inhibit MTCs.

As a caveat, differences in $GABA_AR$ subunit composition may alternatively underlie the observed differences in event kinetics [90]. In the OB, faster α1 subunits distribute equally throughout the EPL while slower α3 subunits concentrate in the upper EPL [90–94], and α1 knockout slows miniature IPSC decay (though not rise times) in MTCs [94]. At least 2 points suggest that the faster rise times of FSI-mediated versus spontaneous inhibition are not due to differential $GABA_AR$ subunit composition, however. First, if faster α1 subunits selectively concentrated postsynaptic to FSIs, then the noted laminar difference in subunit expression would suggest that FSIs preferentially innervate MCs instead of TCs, which we did not observe. Second, differences in subunit expression arise predominantly across cells and not across synapses within the same cell [90,93,94]. Selective localization of faster α1 subunits postsynaptic to FSIs would thus further suggest that FSIs preferentially innervate only subsets of MTCs, which was not supported by the high connectivity rates observed.

While our results suggest that FSIs may preferentially inhibit perisomatic MTC domains, they do not exclude FSI interactions with lateral dendrites of distal MTCs. Indeed, monosynaptic rabies viral tracing has revealed broad connectivity of PV[+] EPL-INs with MTCs, well exceeding MTC–GC connectivity ranges and consistent with broader odor-tuning of PV[+] EPL-INs than GCs [17,95]. Interpretation of these results has further led to the hypothesis that proximal MTC inhibition may be primarily mediated by GCs and not by PV[+] EPL-INs [64,95,96], seemingly in contrast to our present findings. As a caveat, however, the efficiency, directionality, and even specificity with which rabies transmits across dendrodendritic and dendrosomatic synapses in the OB (and how that relates to synaptic strength) remains partially unclear. For example, rabies transmits between GCs and PV[+] EPL-INs [95,97,98] despite no synaptic connections [99]. Moreover, and of particular note, while viral tracing uncovered

broad MTC–PV$^+$ EPL-IN connectivity, the greatest density of connections was still observed among nearby MTCs and PV$^+$ EPL-INs [95], where we have shown that FSIs perisomatically inhibit MTCs. Further, despite GC densities overall far exceeding PV$^+$ EPL-IN densities [5], the number of virally traced short-range MTC–PV$^+$ EPL-IN connections well exceeded the number of short-range MTC–GC connections [95], further suggesting that proximal MTC inhibition may in fact be primarily mediated by PV$^+$ EPL-INs and not GCs. Our present results, revealing prominent perisomatic inhibition of MTCs by nearby FSIs, align well with this latter model. Understanding how such short-range connectivity complements long-range MTC–FSI connectivity stands as an important direction of future research (Fig 8). Given that only approximately 50% of PV$^+$ EPL-IN synapses are reciprocal [40,41], key questions going forward include: (1) how common is distal MTC-to-FSI excitation; (2) how does such distal excitation compare in strength to proximal excitation; and (3) is such distal excitation matched by reciprocal FSI-to-MTC lateral dendrite inhibition? Our simultaneous whole-cell recording approach, combined with precise post hoc morphological reconstruction, is ideally suited to answer these questions.

Despite structural and functional evidence of perisomatic MTC inhibition by FSIs, we failed to detect canonical basket-like innervation patterns using several complementary strategies. Absence of such profuse perisomatic innervation may reflect 2 unique features of MTC–FSI signaling. First, distinct from projection neurons elsewhere in the brain such as neocortex and hippocampus, both somata and dendrites of MTCs are fully excitable and release-competent [100]. Inhibition targeted exclusively to the soma would thus be less effective in controlling MTC signaling than in the case of basket innervation of pyramidal cell somata, through which all synaptic inputs must sum in order to trigger axonal output. Second, distinct from fast-spiking basket cells, FSIs in the OB are anaxonic and inhibit MTCs through dendritic GABA release. Distinct metabolic and trafficking requirements of axons versus dendrites thus limit the total cable length available for FSIs to innervate MTCs, prohibiting profuse somatic wrapping. Moreover, by housing both pre- and postsynaptic machinery, FSI dendrites must distribute in a manner facilitating both GABA release onto MTCs as well as integration of glutamatergic input from the same and other MTCs. By avoiding profuse basket innervation of individual MTC somata, FSI dendrites can likely reciprocally communicate with a larger ensemble of MTCs.

While lacking canonical basket-like structure, perisomatic inhibition of MTCs by FSIs nevertheless proved surprisingly strong. Unitary FSI-to-MTC inhibition hyperpolarized MTCs by $0.46 \pm 0.45$ mV from resting potentials of $-53.0 \pm 3.3$ mV ($n = 8$), comparable to unitary basket cell hyperpolarization of hippocampal and neocortical pyramidal cells by 0.45 and 1.2 mV, respectively, from resting potentials of $-55$ to $-60$ mV in adult rats [101,102] and far exceeding the predicted unitary GC-to-MTC IPSP amplitude of 0.01 to 0.03 mV [57]. FSIs in the OB may thus achieve comparable function as cortical basket cells through strikingly distinct synaptic architecture.

Close proximity between pre- and postsynaptic machinery at reciprocal dendrodendritic and dendrosomatic synapses between FSIs and MTCs further suggests that FSIs may also support parallel subcellular processing, with excitatory input depolarizing nearby active zones to trigger local GABA release in the absence of a global spike. Such local subthreshold GABA release would not only enable FSIs to dynamically shift the balance between recurrent and lateral inhibition among MTCs, but further regulate flexible ensembles of MTCs, dramatically augmenting the computational power of each individual FSI. At least 2 additional factors may further enhance parallel subcellular processing within FSIs. First, FSI expression of philantho-toxin-sensitive Ca$^{2+}$-permeable AMPARs suggests that local FSI release may be dynamically enhanced by coincident Ca$^{2+}$ influx through neighboring AMPARs and Ca$^{2+}$ channels during

MTC–FSI synchronization. Second, multiple dendritic branches of each PV$^+$ EPL-IN strikingly harbor clusters of Na$^+$ channels that colocalize with some features of conventional axon initial segments [48,88], suggesting that FSIs may additionally be capable of generating local dendritic spikes.

## Mechanisms and implications of high-fidelity recurrent and lateral inhibition

Our results show that FSIs drive powerful recurrent and lateral MTC inhibition. That contribution of single FSIs to these signaling modes was not only detectable, but constituted a large fraction of the total inhibition recorded at the soma under our experimental conditions was remarkable given that inhibition onto any MTC should conceivably also reflect the summed input from $10^4$ GCs [56,57] as well as a complex array of glomerular layer circuits [5,11,103]. Our results thus suggest that OB output, specifically reflecting MTC somatic spiking, may be dominantly shaped by the comparatively small population of FSIs. A key component underlying this outsized contribution is the surprising finding that 23% of FSIs responded to unitary MTC release with suprathreshold activation. Such prevalent synaptic detonation under baseline conditions not only distinguishes FSIs from GCs as well as rodent neocortical and hippocampal basket cells [104,105], but is further uncommon throughout the entire rodent brain, typically manifesting at specialized connections crucial for high-fidelilty circuit operation [106] and learning [107,108]. Understanding how the structure, function, and plasticity of MTC–FSI synapses may relate to these specialized detonating connections represents a key area of future investigation.

Other functional differences between FSIs and GCs beyond postsynaptic excitation further motivate reevaluation of key tenets of recurrent and lateral MTC inhibition in the EPL and OB sensory processing overall. In particular, while GC output can be strongly gated by centrifugal cortical input [109], FSIs readily mediate both recurrent and lateral inhibition in acute slices with disrupted corticobulbar communication. FSIs and GCs may concordantly regulate OB sensory processing in context-independent and -dependent forms, respectively, paralleling complementary intrinsic versus extrinsic modulation of hippocampal activity by PV$^+$ versus PV$^-$ basket cells [39]. Alternatively, FSIs may also integrate various cortical and/or neuromodulatory centrifugal input to influence OB sensory processing in a context-dependent manner distinct from that of GCs. Indeed, recent investigation by Fukunaga and colleagues suggests that perisomatic inhibition, as we show here to be a prominent signaling mode in the OB mediated by FSIs, may be pivotal in reward modulation of MC sensory responses [110]. Direct investigation of whether FSIs receive centrifugal cortical and/or neuromodulatory inputs thus stands as an important next step in evaluating contextual modulation of OB sensory processing.

Of further interest, GCs subdivide into superficial- and deep-branching subpopulations selectively innervating TCs and MCs, respectively, to support distinct modes of slow lateral inhibition and contrast enhancement within the 2 projection neuron populations [35]. While we did not directly test whether individual FSIs can innervate both TCs and MCs, the lack of difference in EPL position between FSIs of synaptically coupled TC–FSI and MC–FSI pairs, together with the equally high TC and MC connectivity rates observed across the FSI population, intriguingly suggests that FSIs may not only account for the synchronous spontaneous inhibition observed within nearby TC–MC pairs [111], but may also mediate fast lateral inhibition between TCs and MCs. Such a link, while speculative, could potentially coordinate fast timescale spike-time patterning across the parallel streams of OB output, and strongly motivates future TC–FSI–MC triplet recordings.

Network simulations have demonstrated that high-fidelity perisomatic lateral inhibition, as mediated by FSIs, is likely to promote MTC synchronization [59,60], consistent with mechanisms well established in neocortical and hippocampal circuits [1,2]. This, together with the demonstration that sparse MTC synchronization is sufficient to activate the majority of postsynaptic FSIs, suggests that MTC–FSI interactions may play a lead role in promoting the fast-timescale MTC synchronization driving gamma-frequency OB oscillations. Further experiments are needed to evaluate this hypothesis, however, particularly given the concurrent occurrence of presumably GC-mediated slow lateral inhibition, which itself is capable of promoting synchronization of resonant gamma-frequency MTC spiking [20,63,112]. Integration of the present unitary synaptic and physiological FSI data into advanced biophysical OB simulations (e.g., [113]) will provide a powerful platform for further investigating precisely when and to what degree FSIs synchronize OB output.

## Materials and methods

### Animals

All experiments were completed in compliance with the guidelines established by the Institutional Animal Care and Use Committee of Lehigh University (protocol 1580421). OMP-ChR2 mice were maintained on an albino C57BL/6J background and used as heterozygotes on C57BL/6J or albino C57BL/6J background to minimize olfactory sensory neuron signaling deficits, as previously described [114]. All other electrophysiological experiments used multiple strains of mice on the C57BL/6J background and lacking genetic labeling of interneuron populations, and are thus considered wild type; strains included: M71-IRES-Cre [115], M72-IRES-ChR2:EYFP [116], M72-IRES-tauCherry [117], and Tbet-Cre [118], with no difference in results between strains. Immunohistochemical experiments used identical strains of mice, in addition to a subset of experiments (Fig 2M–T) using compound heterozygous crosses of gene-targeted PV-IRES-Cre mice [119] to either RCL-tdTomato mice [120] or RCL-ReaChR:mCitrine mice [121] maintained on the C57BL/6J background. Mice were socially housed when possible and maintained on a 12 h light/dark cycle with ad libitum access to food and water.

### Slice preparation

Experiments were performed in acute slices prepared from P21-28 mice ($n = 70$), consistent with the full maturation of MTC intrinsic and synaptic properties [122–124] and PV$^+$/CRH-Cre$^+$ EPL-IN densities [125–127]. Mice of both sexes were used. For slice preparation, mice were anesthetized with isoflurane and decapitated into ice-cold oxygenated dissection solution containing the following (in mM): 125 NaCl, 25 glucose, 2.5 KCl, 25 NaHCO$_3$, 1.25 NaH$_2$PO$_4$, 3 MgSO$_4$, and 1 CaCl$_2$. Brains were isolated and acute horizontal slices (310 μm thick) were prepared using a vibratome (VT1200S, Leica Biosystems). Slices recovered for 30 min in approximately 37˚C oxygenated Ringer's solution that was identical to the dissection solution except for lower Mg$^{2+}$ concentrations (1 mM MgSO$_4$) and higher Ca$^{2+}$ concentrations (2 mM CaCl$_2$). Slices were then stored at room temperature until recording. Slices were prepared at approximately the same time each day relative to the animal facility light/dark cycle.

### Electrophysiology

Slices were continuously superfused with warmed oxygenated Ringer's solution (temperature measured in bath: 30 to 32˚C). Tissue was visualized using infrared differential interference contrast (IR-DIC) video microscopy. Recordings were targeted to the medial OB, where the

MCL reliably appears as a uniformly compact cell layer, facilitating the differentiation of cell types. MTC cell types were identified as previously [63,128]. Specifically, MTCs with ≥50% of their soma displaced above the outer edge of the MCL border were classified as TCs; remaining MTCs within the MCL were classified as MCs. TCs with somata still contacting the MCL border were classified as deep TCs (dTCs); those with somata separated from the MCL and within the lower half of the EPL were classified as middle TCs (mTCs); remaining TCs with somata separated from the MCL and within the upper half of the EPL were classified as superficial TCs (sTCs). MTC cell type was assessed and classified during live IR-DIC imaging and further verified in post hoc inspection of intracellular Neurobiotin (NB; Vector Laboratories) or Lucifer Yellow CH (LY; Thermo Fisher Scientific). All recorded MTCs were confirmed to have intact apical dendrites and glomerular tufts (except where noted to exclude potential glomerular layer circuit interactions), as assessed through live imaging, post hoc inspection of intracellular NB or LY, and occurrence of spontaneous long-lasting depolarizations (or associated currents) reflecting glomerulus-wide activation [129,130]. Frequent long-lasting depolarizations led to greater variability in baseline membrane potential in some MTC recordings (e.g., Fig 1M and 1O). For MTC pair and quartet recordings, homotypic versus heterotypic glomerular association was determined by combined assessment of: spontaneous long-lasting depolarization and/or inward current correlation [129,130], comparatively slow lateral excitation among homotypic MCs [62,63], and post hoc inspection of intracellular NB.

Current-clamp data were recorded using electrodes filled with (in mM): 135 K-gluconate 1.8 KCl, 8.8 HEPES, 10 Na-phosphocreatine, 4 Mg-ATP, 0.3 Na-GTP, and 0.2 EGTA. Voltage-clamp data were recorded using electrodes filled with (in mM): 131 CsCl, 3.8 K-gluconate, 0.05 KCl, 8.8 HEPES, 10 Na-phosphocreatine, 4 Mg-ATP, and 0.3 Na-GTP. Electrode solutions additionally contained 8.8 mM GABA or glutamate (for EPL-IN or MTC recordings, respectively) to preclude rundown of dendritic release [54,76,131–133], as well as either 0.025 mM Alexa Fluor 488 or 594 hydrazide to permit live visualization and 0.2% NB to permit post hoc inspection. For a subset of recordings, voltage-clamp data were recorded using electrodes filled with (in mM): 66 CsCl, 61 Cs-gluconate, 3.8 K-gluconate, 0.9 KCl, 8.8 HEPES, 8.8 glutamate, 10 Na-phosphocreatine, 4 Mg-ATP, 0.3 Na-GTP, and 0.1% LY; these recordings were excluded from other voltage-clamp data for comparisons of absolute IPSC amplitudes across cells (S7, S10, and S11 Figs). Electrode resistances were 3 to 10 MΩ. Current-clamped cells were held at their resting membrane potential (i.e., 0 pA holding current); voltage-clamped cells were held at –70 mV. In current-clamp recordings, pipette capacitance was neutralized and series resistance (EPL-IN: 37.0 ± 0.9 MΩ [$n$ = 145]; MTC: 18.8 ± 0.9 MΩ [$n$ = 54]) was compensated using the MultiClamp Bridge Balance operation. In voltage-clamp recordings, series resistance (MTC: 20.5 ± 0.5 MΩ [$n$ = 235]) was not compensated but was monitored continuously to ensure adequate electrode access and recording quality. Current-clamp recordings with unstable and/or depolarized resting membrane potential (EPL-INs: >–50 mV; MTCs: >–45 mV) were abandoned to exclude damaged or otherwise unhealthy cells from our data set. Cell-attached data were recorded prior to obtaining whole-cell access, using the same electrodes as used for subsequent current- and voltage-clamp recordings. Data were low-pass filtered at 4 kHz and digitized at 10 kHz using MultiClamp 700A and 700B amplifiers and an ITC-18 acquisition board controlled by custom software written in IGOR Pro.

To probe unitary synaptic output of EPL-INs onto MTCs, we monitored MTC voltage ($n$ = 54 pairs) or current ($n$ = 91 pairs) while injecting 1-ms suprathreshold current pulses (1.5 to 2.5 nA) into the EPL-IN to trigger single spikes. Voltage-clamped MTCs were recorded with a high-Cl⁻ internal solution to reverse and amplify the GABAergic driving force, enabling IPSCs to be detected at hyperpolarized potentials. In the same pairs, reciprocal unitary synaptic output of the MTC onto the EPL-IN was investigated by monitoring EPL-IN voltage while

injecting a 1-ms suprathreshold current pulse (1.5 to 2.5 nA) or a 2-ms voltage step to 0 mV (approximating the depolarization of a single spike) in the current- or voltage-clamped MTCs, respectively. Synaptic pharmacology was assessed using bath application of DL-AP5 (50 μM; Tocris, 3693), NBQX (10 μM; Tocris, 1262), gabazine (10 μM; Tocris, 1262), and PhTx-74 (10 μM; Alomone Labs, P-120).

To evaluate whether MTCs and FSIs were also linked by electrical coupling, we injected hyperpolarizing step currents sequentially into each cell of a subset of synaptically coupled MTC–FSI pairs recorded in current-clamp mode and calculated coupling coefficients. MTC step current injection ($-407 \pm 64$ pA) strongly hyperpolarized MTC membrane potentials ($\Delta V = -32.9 \pm 2.0$ mV) but caused no significant change in FSI membrane potentials ($\Delta V = -0.02 \pm 0.08$ mV; $p = 0.8$, $t_6 = -0.3$, $n = 7$, $t$ test), yielding an MTC-to-FSI coupling coefficient not significantly different from zero ($0.13 \pm 0.26\%$, $p = 0.7$, $t_6 = 0.5$, $n = 7$, $t$ test). Similarly, FSI step current injection ($-220 \pm 58$ pA) strongly hyperpolarized FSI membrane potentials ($\Delta V = -19.3 \pm 1.4$ mV) but caused no significant change in MTC membrane potentials ($\Delta V = -0.03 \pm 0.09$ mV; $p = 0.7$, $t_4 = -0.4$, $n = 5$, $t$ test), yielding an FSI-to-MTC coupling coefficient also not significantly different from zero ($0.18 \pm 0.47\%$, $p = 0.7$, $t_4 = 0.4$, $n = 5$, $t$ test).

For MTC–FSI pair recordings combined with optogenetic photostimulation, slices were illuminated by a 75 W xenon arc lamp passed through a YFP filter set and 60× water-immersion objective centered on a nearby lateral glomerulus (approximately 2 to 3 glomeruli away from the recorded MTC glomerulus), with field-stop closed to achieve single glomerulus activation, as previously performed [114]. Photostimulation consisted of a 10-ms light pulse.

## Histology

For fluorescent NB labeling with post hoc immunohistochemistry, acute slices containing NB-filled cells were fixed with paraformaldehyde (PFA; 4%) in phosphate buffer (PB; 0.1 M) for >24 h at 4˚C, washed, and then incubated in blocking solution (2% normal serum and 0.4% Triton X-100, in PB). Slices were then incubated in blocking solution containing Alexa Fluor 488 or 594 Streptavidin (1 μg/ml) and primary antibodies, washed, incubated in blocking solution containing secondary antibodies and Hoechst 33342 (0.25 μg/ml), washed, and mounted with Fluoromount G (SouthernBiotech). For fluorescent NB labeling without post hoc immunohistochemistry, antibodies were omitted. Non-fluorescent chromogenic NB labeling was performed as previously described [71].

For immunohistochemistry, adult mice were anesthetized with intraperitoneal injection of ketamine (200 mg/kg) and xylazine (20 mg/kg) and then transcardially perfused with 0.1 M phosphate-buffered saline followed by PFA (4%) in PB. Brains were extracted, postfixed overnight, and then sectioned with a vibratome (Leica, VT1000S). Free-floating 50-μm sagittal sections were then incubated in blocking solution (2% normal serum and 0.1% Triton X-100, in PB), washed, and incubated in antibody solution (2% normal serum and 0.05% Tween 20, in PB) containing combinations of the following primary antibodies: guinea pig anti-parvalbumin (1:2,000; Synaptic Systems, 195 004), mouse anti-synaptotagmin-2 (1:200; Zebrafish International Resource Center, znp-1), rabbit anti-TRIM-46 (1:1,000; Synaptic Systems, 377 008), rabbit anti-Lucifer Yellow (1:1,000; Thermo Fisher Scientific, A-5750), rabbit anti-parvalbumin (1:2,000; Synaptic Systems, 195 002), rabbit anti-synaptobrevin-1 (1:500; Synaptic Systems, 104 002), rabbit anti-vasoactive intestinal peptide (1:4,000; ImmunoStar, 20077), and sheep anti-tyrosine hydroxylase (1:1,000; Millipore, AB1542). Sections were then washed, incubated in antibody solution with fluorescent secondary antibodies and Hoechst 33342, washed, and mounted with Fluoromount G. All washes were performed with PB. All incubation steps took place for 1 to 3 h at room temperature or overnight at 4˚C. Secondary

antibodies were used at 1:600. At least 2 sections from each of 2 male and 2 female mice were examined for each experiment; no gross sex-dependent differences were observed.

Brightfield and fluorescent widefield images were collected on an upright Nikon Eclipse E1000 microscope using 20× air and 100× oil-immersion objectives. Fluorescent confocal z-stacks were collected on an inverted Zeiss LSM 880 confocal microscope using a 20× air objective and 25× and 63× oil-immersion objectives, 0.5-μm z-steps, 2,048 × 2,048 resolution, and 3-pixel median filter to reduce noise. Morphological images included in figures show maximum- or minimum-intensity projections of fluorescent confocal z-stacks or chromogenic widefield z-stacks, respectively, except where noted. Neuron morphologies were reconstructed from confocal z-stacks and analyzed using the SNT plugin in ImageJ (Fiji) [134]. Intersomatic distances represent the shortest distance between somata irrespective of direction (zero for directly contacting somata) and were measured from single widefield fluorescent or brightfield images of intracellular NB and LY, and thus do not account for potential differences in cell depth in the tissue.

## Data analysis

Values reported are either mean ± SEM or median (first quartile [$Q_1$], third quartile [$Q_3$]) for normally or non-normally distributed data, respectively. Line plots with thin and thick lines denote individual trials and mean, respectively. Line plots with shading denote mean ± SEM. Single, double, and triple asterisks in figures and tables denote statistical significance at $p < 0.05$, $p < 0.01$, and $p < 0.001$ levels, respectively. For each statistical test, data normality was first determined by the Shapiro–Wilk test, and nonparametric tests applied where appropriate. For visual comparison of normally distributed data, all individual data points are displayed in addition to sample mean and SEM. For visual comparison of non-normally distributed data, data are displayed as standard boxplots, with data points denoting sample outliers. Significance thresholds were corrected for multiple comparisons where indicated. All recorded traces, spikes, and potential unitary synaptic events were visually inspected for detection accuracy. No differences in synaptic connectivity or intrinsic biophysical properties were detected between sex, and data were therefore pooled across male and female mice.

EPL-INs were classified as FSIs or RSIs using step current-evoked spiking responses, as shown in Fig 1B–1E and reinforced by unbiased agglomerative hierarchical clustering (Fig 1F) and principal component analysis (S1 Fig). Hierarchical clustering was performed on z-scored intrinsic biophysical properties using Ward's method, with properties ordered as in S1 Fig, beginning with AHP 50% decay. Given the binary connectivity profile observed among MTCs and FSIs versus RSIs, for analysis of unitary synaptic properties the presence of a unitary connection was used as a complementary criterion to identify an additional 10 FSIs from pair recordings for which step current-evoked spiking responses were not obtained. Results did not differ if these 10 FSIs were omitted.

Resting membrane potential was recorded immediately after obtaining whole-cell access, and for EPL-INs was defined as the 10th percentile of voltage recorded for each cell to limit the contribution of prodigious sEPSP rates (e.g., S2 Fig). Membrane time constant, input resistance, and capacitance were calculated using the voltage trajectory and maximum voltage change evoked by a 50-pA, 100-ms hyperpolarizing step current injection (e.g., S2 Fig). Spontaneous firing rates were defined as the spike-count firing rate calculated from the total number of spikes not driven by evoked unitary synaptic input recorded over 151.6 s (103.2, 184.0) ($n = 145$) for each cell.

Firing rate-current (FI) curves were examined using 500-ms depolarizing step current injections ranging from 50 to 600 pA in steps of 50 pA for FSIs and less excitable RSIs, while

RSIs that readily underwent depolarization block were examined using step currents ranging from 10 to 100 pA in steps of 10 pA (e.g., S2 Fig). FI curve spike times were detected using a voltage derivative threshold of 15 mV/ms. FI curve rates were calculated from the median inverse interspike interval (ISI) evoked by each step current. Spike properties (with the exception of afterdepolarization [ADP] amplitudes) were calculated from the first spike evoked by the weakest suprathreshold step current (i.e., rheobase). Spike amplitude was calculated as the difference between spike threshold (i.e., the voltage at spike onset) and spike peak. Maximum and minimum spike slopes were calculated as the maximum and minimum voltage derivatives. Spike width was calculated as the full-width at half-maximum spike amplitude. Spike afterhyperpolarization (AHP) amplitude was calculated as the difference in spike threshold and the minimum voltage reached within 10 ms of spike onset. AHP 50% decay was calculated as the latency from AHP onset (i.e., spike falling phase matching spike threshold) to decay of the AHP to 50% of its amplitude. Maximum FI curve gain was calculated as the maximum FI curve derivative. Maximum FI curve rate was calculated as the inverse of the minimum ISI detected. Relative and absolute spiking adaptation were calculated from the response to the weakest suprathreshold step current evoking sustained activity ($\geq 5$ spikes) and from the first evoked spike cluster within that response for FSIs exhibiting clustered spiking (e.g., Fig 1B and 1C).

Spike times evoked by 1-ms suprathreshold current pulses (1.5 to 2.5 nA) or unitary MTC release were calculated as the time at which membrane potentials (upsampled 100-fold) exceeded –30 mV. EPL-IN spike waveforms evoked by such current pulses typically lacked an AHP and instead displayed an ADP (e.g., S2 Fig). ADP amplitudes were calculated as the difference between the minimum and maximum post-spike voltage (occurring within 6 ms of spike falling phases) for spike waveforms lacking an AHP.

Postsynaptic events were detected using a standard template-matching function in Axograph [135] with double-exponential template (S3 Table). For analysis of unitary MTC–EPL-IN synaptic connectivity, presynaptic activation (either pulse-evoked spikes or brief voltage-steps) was triggered every 8 to 22 s and peristimulus time histograms of postsynaptic events were calculated across all trials. Unitary connectivity was classified as significant if the probability of a postsynaptic event within the first time bin following presynaptic spike time or voltage step onset exceeded the mean + the standard deviation × multiplication factor of the event probability across the 1 s preceding presynaptic spike time or voltage step onset. For tests of unitary EPL-IN excitation, 5 ms time bins were used (accommodating longer latencies following presynaptic voltage steps—see below) and the significance multiplication factor was set to 2.75; suprathreshold postsynaptic responses occurring within 5 ms of presynaptic activation (i.e., detonation) were additionally used to classify a unitary connection as significant. For tests of unitary MTC inhibition, 2.5 ms time bins were used and the significance multiplication factor was set to 3. Unitary postsynaptic event latency was calculated as the latency from presynaptic spike onset to the time at which the postsynaptic response (upsampled 20-fold) reached 5% of its amplitude. Unitary FSI excitation latencies were significantly longer ($2.36 \pm 0.06$ ms versus $1.17 \pm 0.04$ ms; $p = 6.6 \times 10^{-25}$, $t_{75} = 15.4$, $t$ test) and had higher jitter (0.47 [0.35, 0.62] versus 0.24 [0.12, 0.26]; $p = 2.7 \times 10^{-7}$, r.s. = 2,290, Wilcoxon rank-sum test) when measured from MTC depolarization onset in voltage-clamp mode ($n = 46$) than from MTC spike onset in current-clamp mode ($n = 31$), reflecting lower precision in identifying MTC activation timing in voltage-clamp mode. All unitary FSI excitation latencies reported in the main text, including uEPSP and detonation latencies, are thus restricted to pairs recorded in current-clamp mode with precisely measured presynaptic spike onsets. Postsynaptic rise times were calculated as the duration for the postsynaptic response (upsampled 20-fold) to increase from 20% to 80% of its amplitude. sEPSP rates were calculated for each EPL-IN from the total number of EPSPs occurring in the 1 s preceding presynaptic activation across all

trials, encompassing 39.9 ± 1.3 s ($n$ = 145) total recording for each cell. Median sEPSP half-widths were calculated as the full-width at half-maximum amplitude of the median sEPSP waveform. For FSIs exhibiting an uEPSP in MTC–FSI recordings, $\Delta V_{thresh.}$ was calculated as the difference in rheobase spike threshold and median membrane potential prior to unitary excitatory input.

For analysis of cell-attached spontaneous spike-time synchrony (Fig 3C), spike times were detected using a current threshold of 18 pA and then convolved with a Gaussian kernel (1 ms standard deviation) to account for slight differences in spike waveforms. Trains of convolved spike times were then mean-subtracted and the cross-correlogram calculated.

For analysis of lateral inhibition among MTCs, peristimulus time histogram analysis was used as above to classify lateral inhibition as significant, with 10 ms time bins (accommodating disynaptic latencies) and a significance multiplication factor of 2 for current-clamped MTCs and 3 for voltage-clamped MTCs.

## Supporting information

**S1 Fig. Principal component analysis reinforces subdivision of EPL-INs into FSIs and RSIs.** (**A**) Projection of 104 EPL-INs onto the first 2 principal components (PC1 and PC2) defined by principal component analysis of z-scored intrinsic biophysical properties, revealing 2 major clusters matching FSI (green) and RSI (orange) subtypes. (**B**) Decomposition of PC1 and PC2 loading by each intrinsic biophysical property. Source data for panels A and B are provided in Supporting information, S7 Data.
(TIF)

**S2 Fig. EPL-IN subtypes exhibit stark differences in most intrinsic biophysical properties.** (**A–E**) Diverse responses used to calculate intrinsic biophysical properties for the example FSI from Fig 1B, including: spontaneous activity at resting membrane potential (**A**), mean response to negative step current injection, with single-exponential fit (dashed black line) (**B**), mean spike waveform evoked by 1-ms suprathreshold current injection (**C**), firing rate-current relationship (**D**), and interspike interval (ISI) coefficient of variation evoked by positive step current injection (**E**). Spontaneous spike in **A** truncated to better visualize synaptic activity. Inset in **A**: enlargement of boxed region. **F–T:** Same as **A–E** for the example FSI and RSIs from Fig 1C–1E. Insets in **A, F, K, P** are identically scaled. Source data for panels D, E, I, J, N, O, S, and T are provided in Supporting information, S8 Data.
(TIF)

**S3 Fig. PV expression distinguishes FSIs from RSIs.** Intracellular NB and post hoc PV staining with 50-μm magnified region centered on somata (left), inverted NB (middle), and step current-evoked spiking (right) of a panel of EPL-INs. Spiking responses are color-coded to reflect FSI vs. RSI physiology, as in Fig 1.
(TIF)

**S4 Fig. Neither FSIs nor RSIs are dopaminergic.** Intracellular NB and post hoc TH staining with 50-μm magnified region centered on somata (left), inverted NB (middle), and step current-evoked spiking (right) of a panel of EPL-INs. Spiking responses are color-coded to reflect FSI vs. RSI physiology, as in Fig 1. An example TH[+] short-axon cell (SAC) exhibiting tonic spontaneous firing is additionally included as positive control for TH staining.
(TIF)

**S5 Fig. VIP expression poorly distinguishes FSIs and RSIs.** Intracellular NB and post hoc VIP staining with 50-μm magnified region centered on somata (left; single optical confocal

planes), inverted NB (middle; maximum-intensity confocal projection), and step current-evoked spiking (right) of a panel of EPL-INs. Spiking responses are color-coded to reflect FSI vs. RSI physiology, as in Fig 1.
(TIF)

**S6 Fig. Unitary MTC-to-RSI excitation in a solitary example was distinctly weaker than FSI excitation.** (**A, B**) Step current-evoked spiking response (**A**) and unitary synaptic interactions (**B**) for the solitary MTC–RSI pair exhibiting significant unitary MTC-to-RSI excitation (morphology not recovered). Asterisk marks significant postsynaptic response. Inset: mean postsynaptic RSI voltage. (**C**) The MTC-to-RSI uEPSP amplitude was markedly weaker than FSI uEPSPs ($n = 69$). Source data for panel C are provided in Supporting information, S9 Data.
(TIF)

**S7 Fig. Connected MTC–FSI pairs exhibit shorter intersomatic distances than unconnected pairs.** (**A**) MTC–RSI pairs ($n = 19$) exhibited modestly shorter intersomatic distances than MTC–FSI pairs ($n = 97$) (*$p = 0.02$, r.s. = 808, Wilcoxon rank-sum test). (**B**) Among MTC–FSI pairs, reciprocally connected pairs exhibited shorter intersomatic distances than unconnected pairs ($p = 0.02$, $F_{3,55} = 3.4$, one-way ANOVA; reciprocal ($n = 32$) vs. excitation only ($n = 9$): $p = 0.3$, reciprocal vs. inhibition only ($n = 4$): $p = 0.4$, reciprocal vs. unconnected ($n = 14$): $p = 0.03$, excitation only vs. inhibition only: $p = 1.0$, excitation only vs. unconnected: $p = 0.9$, inhibition only vs. unconnected: $p = 1.0$, post hoc Tukey–Kramer test). (**C**) MTC–FSI pairs with significant unitary MTC-to-FSI excitation ($n = 71$) exhibited shorter intersomatic distances than pairs with no excitatory connectivity ($n = 26$) (**$p = 2.5 \times 10^{-3}$, r.s. = 3,108, Wilcoxon rank-sum test). (**D**) MTC–FSI pairs with significant unitary FSI-to-MTC inhibition ($n = 36$) exhibited shorter intersomatic distances than pairs with no inhibitory connectivity ($n = 23$) (**$p = 9.7 \times 10^{-3}$, $t_{57} = 2.7$, two-sample $t$ test). Analysis restricted to pairs with voltage-clamped MTCs (and therefore sensitive detection of unitary inhibition). (**E, F**) Neither MTC-to-FSI uEPSP amplitudes (**E**) nor FSI-to-MTC uIPSC amplitudes (**F**) correlated with intersomatic distance (uEPSP: $n = 77$; $p = 0.1$, $t_{75} = 1.7$, linear regression, slope not significantly different from 0; uIPSC: $n = 29$; $p = 0.4$, $t_{27} = 0.8$, linear regression, slope not significantly different from 0). Pairs lacking connectivity (i.e., uEPSP or uIPSC amplitude of zero) not included in analysis. Source data for all panels are provided in Supporting information, S10 Data.
(TIF)

**S8 Fig. Unitary MTC–FSI synaptic pharmacology.** (**A**) Recording from an example MTC–FSI pair (morphology not recovered) showing MTC-to-FSI uEPSP amplitudes before and after combined bath application of glutamatergic antagonists NBQX (10 μM) and AP5 (50 μM) and subsequent application of GABA$_A$R antagonist gabazine (10 μM). (**B**) Postsynaptic FSI voltages from the pair in **A**. Traces in each subplot correspond to the bracketed trials in **A**. Asterisk marks significant unitary postsynaptic response measured during control trials. (**C**) Unitary MTC-to-FSI excitation was blocked by combined application of NBQX and AP5 and partially recovered upon wash-out in 4 MTC–FSI pairs ($p = 2.2 \times 10^{-12}$, $F_{3,12} = 414.9$, one-way ANOVA; ctrl vs. N/A: ***$p = 5.5 \times 10^{-9}$, ctrl vs. N/A/G: ***$p = 5.5 \times 10^{-9}$, ctrl vs. wash: ***$p = 5.5 \times 10^{-9}$, N/A vs. N/A/G: $p = 0.9$, N/A vs. wash: *$p = 0.03$, N/A/G vs. wash: **$p = 0.01$, post hoc Tukey–Kramer test). (**D, E**) Same as **A and B** for FSI-to-MTC uIPSCs recorded in the same example pair. (**F**) Unitary FSI-to-MTC inhibition was blocked by application of gabazine and partially recovered upon wash-out in the same 4 MTC–FSI pairs as **C** ($p = 1.7 \times 10^{-3}$, $F_{3,12} = 9.5$, one-way ANOVA; ctrl vs. N/A: $p = 0.33$, ctrl vs. N/A/G: **$p = 2.1 \times 10^{-3}$, ctrl vs. wash: **$p = 8.3 \times 10^{-3}$, N/A vs. N/A/G: *$p = 0.046$, N/A vs. wash: $p = 0.17$, N/A/G vs. wash:

$p$ = 0.85, post hoc Tukey–Kramer test). Filled symbols in **C and F** correspond to the example pair shown. Source data for panels A, C, D, and F are provided in Supporting information, S11 Data.
(TIF)

**S9 Fig. MTC–FSI connectivity is restricted to infraglomerular layers.** (**A**) Example MTC–FSI pair with MTC apical dendrite truncated prior to entering glomerular layer (open arrowhead). (**B, C**) FSI fast-spiking response to step current injection (**B**) and unitary synaptic connectivity with MTC (**C**). Asterisks mark significant unitary postsynaptic responses. (**D–F**) Same as **A–C** for a second example MTC–FSI pair.
(TIF)

**S10 Fig. MCs and TCs exhibit similar unitary connectivity with FSIs.** (**A**) Detection of unitary FSI excitation did not significantly differ between MC–FSI and TC–FSI pairs (upper; $p$ = 0.3, $\chi^2_{[1]}$ = 1.0, $\chi^2$ test) even when considering only middle and superficial TCs (m/sTCs) to exclude potential misclassification of deep TCs (lower; $p$ = 0.2, $\chi^2_{[1]}$ = 1.3, $\chi^2$ test). Likewise, the proportion of FSIs responding to unitary MTC release with detonation did not significantly differ between MC–FSI and TC–FSI pairs (upper; $p$ = 0.3, $\chi^2_{[1]}$ = 1.2, $\chi^2$ test) or between MC–FSI and m/sTC–FSI pairs (lower; $p$ = 0.1, $\chi^2_{[1]}$ = 2.3, $\chi^2$ test). (**B**) FSI uEPSP amplitudes did not significantly differ between MC-FSI ($n$ = 21) and TC-FSI pairs ($n$ = 56) (upper; $p$ = 1.0, r.s. = 820, Wilcoxon rank-sum test) or between MC–FSI and m/sTC–FSI pairs ($n$ = 46) (lower, $p$ = 1.0, r.s. = 711, Wilcoxon rank-sum test). (**C**) Across MTC–FSI pairs with significant unitary MTC-to-FSI excitation, FSI somatic depth throughout the EPL did not differ between MC–FSI ($n$ = 19) and TC–FSI pairs ($n$ = 53) (upper; $p$ = 0.5, r.s. = 646, Wilcoxon rank-sum test) or between MC–FSI and m/sTC–FSI pairs ($n$ = 43) (lower, $p$ = 0.1, $t_{60}$ = 1.5, two-sample $t$ test). (**D**) Detection of unitary MTC inhibition did not significantly differ between MC–FSI and TC–FSI pairs (upper; $p$ = 0.2, $\chi^2_{[1]}$ = 1.9, $\chi^2$ test) or between MC–FSI and m/sTC–FSI pairs (lower; $p$ = 0.1, $\chi^2_{[1]}$ = 2.4, $\chi^2$ test); only voltage-clamped MTCs were considered for peak detection sensitivity. (**E**) MTC uIPSC amplitudes did not significantly differ between MC–FSI ($n$ = 5) and TC–FSI pairs ($n$ = 29) (upper; $p$ = 0.6, r.s. = 100, Wilcoxon rank-sum test) or between MC–FSI and m/sTC–FSI pairs ($n$ = 26) (lower; $p$ = 0.6, r.s. = 90, Wilcoxon rank-sum test). (**F**) Across MTC–FSI pairs with significant unitary FSI-to-MTC inhibition, FSI somatic depth throughout the EPL did not differ between MC–FSI ($n$ = 11) and TC–FSI pairs ($n$ = 33) ($p$ = 0.8, $t_{42}$ = 0.3, two-sample $t$ test) or between MC–FSI and m/sTC–FSI pairs ($n$ = 30) (lower, $p$ = 0.9, $t_{39}$ = 0.2, two-sample $t$ test). (**G**) The proportion of MTC–FSI pairs exhibiting reciprocal unitary connectivity did not differ between MC–FSI and TC–FSI pairs (upper; $p$ = 0.3, $\chi^2_{[1]}$ =1.0, $\chi^2$ test) or between MC–FSI and m/sTC–FSI pairs (lower; $p$ = 0.2, $\chi^2_{[1]}$ = 1.4, $\chi^2$ test). Source data for all panels are provided in Supporting information, S12 Data.
(TIF)

**S11 Fig. Comparison of unitary FSI-to-MTC and MTC-to-FSI synaptic transmission properties.** (**A**) FSI-to-MTC uIPSC latency ($n$ = 44) was significantly shorter than MTC-to-FSI uEPSP latency ($n$ = 31) (***$p$ = 8.0 × $10^{-5}$, r.s. = 1,545, Wilcoxon rank-sum test). (**B**) FSI-to-MTC uIPSC jitter and MTC-to-FSI uEPSP jitter were equivalent ($p$ = 0.5, r.s. = 1,117, Wilcoxon rank-sum test). (**C**) Trial-to-trial FSI-to-MTC uIPSC event probability ($n$ = 44) was significantly lower than trial-to-trial MTC-to-FSI uEPSP event probability ($n$ = 79) (***$p$ = 5.4 × $10^{-8}$, r.s. = 5,838, Wilcoxon rank-sum test). Unitary FSI-to-MTC IPSP latency, jitter, and probability not included in comparisons due to limited unitary IPSP detection sensitivity (Fig 1R). (**D**) Across all MTC–FSI pairs with at least 1 direction of unitary connectivity, FSI-to-MTC uIPSC amplitude positively correlated with MTC-to-FSI uEPSP amplitude

($n = 41$; **$p = 1.9 \times 10^{-3}$, $t_{39} = 3.3$, $R^2 = 0.22$, linear regression, slope significantly different from 0). For pairs exhibiting exclusive FSI detonation, uEPSP amplitudes were estimated as the difference between resting membrane potential and spike threshold ($\Delta V_{thresh.}$), as in Fig 3G. Shading denotes 95% confidence interval. Source data for all panels are provided in Supporting information, S13 Data.
(TIF)

**S12 Fig. Syt2 clusters do not selectively target MTC axon initial segments.** Single confocal optical plane (left) and maximum-intensity projection (right; approximately 50 μm depth) of Syt2 and axon initial segment component TRIM-46 in the OB, revealing an absence of clear Chandelier-like innervation of MTCs.
(TIF)

**S13 Fig. Comparison of spontaneously alternating FSI detonation vs. uEPSP trials reveals high-fidelity recurrent MTC inhibition.** (**A, B**) Example MTC–FSI pair (**A**) in which unitary MTC release triggers FSI detonation on some trials (light green) and uEPSPs on other trials (dark green) (**B**). Asterisk marks significant unitary postsynaptic response. (**C**) Subtraction of mean MTC currents across FSI detonation vs. uEPSP trials from **B** isolates IPSC waveforms time-locked to FSI detonation. (**D–F**) Same as **A–C** for an example MTC-FSI pair recorded in current-clamp, revealing isolation of an IPSP waveform time-locked to FSI detonation.
(TIF)

**S14 Fig. Directly neighboring heterotypic MTCs are preferentially linked by fast lateral inhibition.** (**A**) Among heterotypic MTC pairs tested for both fast and slow lateral inhibition (Fig 5), pairs with fast lateral inhibition ($n = 9$) exhibited shorter intersomatic distances than pairs without fast lateral inhibition ($n = 34$) (**$p = 8.3 \times 10^{-3}$, r.s. = 111, Wilcoxon rank-sum test) (upper). No difference was observed in the number of glomeruli separating the apical dendrites of MTCs with vs. without fast lateral inhibition ($p = 0.4$, $\chi^2_{[2]} = 1.6$, $\chi^2$ test) (lower). (**B**) Pairs with slow lateral inhibition ($n = 9$) exhibited comparable intersomatic distances as pairs without slow lateral inhibition ($n = 34$) ($p = 0.3$, r.s. = 166, Wilcoxon rank-sum test) (upper). No difference was observed in glomerular separation ($p = 0.2$, $\chi^2_{[2]} = 3.4$, $\chi^2$ test) (lower). (**C**) Pairs with fast lateral inhibition ($n = 9$) exhibited comparable intersomatic distances as pairs with slow lateral inhibition ($n = 9$) ($p = 0.2$, r.s. = 70.5, Wilcoxon rank-sum test) (upper). No difference was observed in glomerular separation ($p = 0.1$, $\chi^2_{[2]} = 2.5$, $\chi^2$ test) (lower). (**D, E**) Same as **A and C**, but including heterotypic MTC pairs tested for both fast and slow lateral inhibition (Fig 5) as well as heterotypic MTC pairs comprising each quartet tested for fast lateral inhibition (Fig 6; only single presynaptic MTC activation results included). Pairs with fast lateral inhibition ($n = 17$) exhibited shorter intersomatic distances than pairs without fast lateral inhibition ($n = 60$) (**$p = 9.6 \times 10^{-3}$, r.s. = 453.5, Wilcoxon rank-sum test) (**D** upper) but comparable intersomatic distances as pairs with slow lateral inhibition ($n = 9$) ($p = 0.9$, r.s. = 227.5, Wilcoxon rank-sum test) (**E** upper); no difference was observed in glomerular separation as a function of fast lateral inhibition ($p = 0.07$, $\chi^2_{[2]} = 5.3$, $\chi^2$ test) (**D** lower) or fast vs. slow lateral inhibition ($p = 0.07$, $\chi^2_{[2]} = 3.6$, $\chi^2$ test) (**E** lower). (**F**) Probability of detecting fast lateral inhibition (upper; pairs from Figs 5 and 6) or slow lateral inhibition (lower; pairs form Fig 5) plotted as a function of intersomatic distance starting with 0 μm and then binned in 20-μm increments (exclusive lower bound and inclusive upper bound). Number of pairs in each bin noted next to each marker. Source data for all panels are provided in Supporting information, S14 Data.
(TIF)

**S1 Table. EPL-IN intrinsic biophysical properties.** Source data provided in Supporting information, S15 Data.
(DOCX)

**S2 Table. EPL-IN anatomical and morphometric properties.** Source data provided in Supporting information, S16 Data.
(DOCX)

**S3 Table. Template parameters for postsynaptic event detection.**
(DOCX)

**S1 Data. Source data for Fig 1.** Spreadsheet of data for hierarchical clustering, Sholl analysis, and unitary synaptic connectivity analysis.
(XLS)

**S2 Data. Source data for Fig 2.** Spreadsheet of data for synaptic event rise time analysis.
(XLS)

**S3 Data. Source data for Fig 3.** Spreadsheet of data for unitary FSI excitation and threshold presynaptic MTC population size analysis.
(XLS)

**S4 Data. Source data for Fig 4.** Spreadsheet of data for fast lateral inhibition prevalence and PhTx-74 sensitivity analysis.
(XLS)

**S5 Data. Source data for Fig 5.** Spreadsheet of data for fast and slow lateral inhibition prevalence analysis.
(XLS)

**S6 Data. Source data for Fig 6.** Spreadsheet of data for MTC quartet fast lateral inhibition amplitude and prevalence analysis.
(XLS)

**S7 Data. Source data for S1 Fig.** Spreadsheet of data for principal component analysis.
(XLS)

**S8 Data. Source data for S2 Fig.** Spreadsheet of data for example firing rate-current and firing irregularity-current plots.
(XLS)

**S9 Data. Source data for S6 Fig.** Spreadsheet of data for comparing unitary excitation strength between FSIs and the singularly excited RSI.
(XLS)

**S10 Data. Source data for S7 Fig.** Spreadsheet of data for analysis of unitary connectivity as a function of intersomatic distance.
(XLS)

**S11 Data. Source data for S8 Fig.** Spreadsheet of data for unitary synaptic pharmacology analysis.
(XLS)

**S12 Data. Source data for S10 Fig.** Spreadsheet of data for analysis of unitary connectivity as a function of MTC subtype.
(XLS)

**S13 Data. Source data for S11 Fig.** Spreadsheet of data for comparing unitary FSI excitation and MTC inhibition.
(XLS)

**S14 Data. Source data for S14 Fig.** Spreadsheet of data for analysis of MTC lateral inhibition as a function of intersomatic distance and glomerular separation.
(XLS)

**S15 Data. Source data for S1 Table.** Spreadsheet of data for EPL-IN intrinsic biophysical property analysis.
(XLS)

**S16 Data. Source data for S2 Table.** Spreadsheet of data for EPL-IN anatomical and morphometric property analysis.
(XLS)

## Acknowledgments

We thank members of the Cheetham and Haas laboratories for helpful discussion.

## Author Contributions

**Conceptualization:** Shawn D. Burton, Nathaniel N. Urban.

**Data curation:** Shawn D. Burton, Christina M. Malyshko.

**Formal analysis:** Shawn D. Burton, Christina M. Malyshko, Nathaniel N. Urban.

**Funding acquisition:** Shawn D. Burton, Nathaniel N. Urban.

**Investigation:** Shawn D. Burton, Christina M. Malyshko.

**Project administration:** Shawn D. Burton.

**Supervision:** Shawn D. Burton, Nathaniel N. Urban.

**Writing – original draft:** Shawn D. Burton, Christina M. Malyshko, Nathaniel N. Urban.

**Writing – review & editing:** Shawn D. Burton, Christina M. Malyshko, Nathaniel N. Urban.

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
