## [Editor Report · Decision Letter 0]

3 May 2024

Dear Dr Burton, 

Thank you for submitting your manuscript entitled "Fast-spiking interneuron detonation drives high-fidelity inhibition in the olfactory bulb" for consideration as a Research Article by PLOS Biology.

Your manuscript has now been evaluated by the PLOS Biology editorial staff as well as by an academic editor with relevant expertise and I am writing to let you know that we would like to send your submission out for external peer review.

Once your full submission is complete, your paper will undergo a series of checks in preparation for peer review. After your manuscript has passed the checks it will be sent out for review. To provide the metadata for your submission, please Login to Editorial Manager (https://www.editorialmanager.com/pbiology) within two working days, i.e. by May 05 2024 11:59PM.

Kind regards,

Christian

Christian Schnell, PhD

Senior Editor

PLOS Biology

cschnell@plos.org

---

## [Decision Letter · Decision Letter 1]

13 Jun 2024

Dear Dr Burton,

Thank you for your patience while your manuscript "Fast-spiking interneuron detonation drives high-fidelity inhibition in the olfactory bulb" went through peer-review at PLOS Biology. Your manuscript has now been evaluated by the PLOS Biology editors, an Academic Editor with relevant expertise, and by several independent reviewers.

In light of the reviews, which you will find at the end of this email, we are pleased to offer you the opportunity to address the comments from the reviewers in a revision that we anticipate should not take you very long. We will then assess your revised manuscript and your response to the reviewers' comments with our Academic Editor aiming to avoid further rounds of peer-review, although might need to consult with the reviewers, depending on the nature of the revisions.

**IMPORTANT - SUBMITTING YOUR REVISION**

*Resubmission Checklist*

*Published Peer Review*

*PLOS Data Policy*

*Blot and Gel Data Policy*

Sincerely,

Christian

Christian Schnell, PhD

Senior Editor

PLOS Biology

cschnell@plos.org

REVIEWS:

Reviewer #1: Review of PBIOLOGY-D-24-01303R1, "Fast-spiking interneuron detonation drives high-fidelity inhibition in the olfactory bulb." This study by Burton and Colleagues uses whole-cell patch-clamp physiology and some immunohistochemistry to identify the source of temporally fast and precise inhibition of olfactory bulb output neurons. The study is noted for its extremely detailed and careful electrophysiology, which appears to resolve the conflict between previously observed M/T cell synchrony and the temporally distributed nature of inhibition they receive from granule cells. The experiments are technically demanding, and the results are convincing. The attention to detail and thoroughness are commendable; however, this is also somewhat of a weakness of the manuscript. The details and figures are rather dense, and, in several places, the overall message becomes obscured. This does not dampen my enthusiasm for the study. Still, I recommend that the authors revise their text to make it more accessible for general biologists (PLOS Biology) and not only those with a deep understanding of electrophysiological methods. Overall, this is a strong study with a remarkably detailed investigation. The authors should be able to address each of the points below without the need for further experimentation. 

1) A circuit schematic to accompany the introduction would be helpful and serve as a touchstone for the rest of the study. As a related note, the schematics used in Figure 1S-U are helpful and reappear throughout the study, but a legend or description of what the shapes, colors, and lines represent would be helpful. 

2) In Figure 1 part P/V, why are the voltage recordings from the MT cells so much more variable than those from the FSI or RSIs? Is this meaningful in some way? Can the axes be scaled similarly to the pannels on the left?

3) Line 202: the conceptual jump to cortex/hippocampus is abrupt and needs more of a transition. The authors are encouraged to explain why looking at cortical/hippocampal basket cells is analogous to looking at FSIs in the bulb. The following sentence approaches to explain the connection but is insufficient to establish the concept. "To investigate whether the evidence of perisomatic MTC innervation by single PV+ FSIs presented above collectively manifests in similar canonical patterns of basket-like innervation," Why would one expect to see patterns in the OB similar to the cortex/hippocampus? Is there previous evidence to suggest similar structures to the cortex/ hippocampus exist in the OB?

4) Line 344: typo "enhance"

5) Line 281: "Strikingly, our subtraction procedure indeed isolated clear recurrent IPSCs strongly timelocked to FSI detonation latencies (Fig 3E)." Please explain in sufficient detail what the "subtraction procedure" involves.

6) While the methods section provides sufficient detail, the results section should have some more text on the optogenetic experimental design (i.e. what mice and how was stimulation achieved). 

7) The legend of Figure 7 does not patch part A. The legend states MTC-FSI pair, while the panel A label is "sTC-FSI." This type of nomenclature mismatch comes up in several of the figures. While I understand the authors are using precise descriptors of TC cell types (dTC, mTC, sTC), the distinctions are confusing when the results are generalized to MTCs. Unless there are differences found between the TC subtypes with respect to fast inhibition, I recommend simplifying the nomenclature. 

Overall, this is an excellent study and will be of interest to a wide range of systems neurobiologists. 

Reviewer #2: This work systemically characterized the interneurons particularly the fast spiking (FSIs) type expressing parvalbumin (PV) in the external plexiform layer (EPL) of the olfactory bulb and their functional impact on output neurons with anatomical and physiological approaches. The main conclusion is that the EPL-FSIs powerfully regulate OB output via their perisomatic synapses with OB output neurons MTCs. Although morphology and physiology of the PV-expressing OB EPL interneurons were reported by a several studies, this study revealed new features of these inhibitory INs including the noncanonical perisomatic detonation, advanced our understanding their functional roles in mediating fast lateral inhibition, is especially of interest to the field of olfactory neuroscience. The manuscript is well organized with most conclusions appropriately drawn based on evidence from well-designed experiments. The Methodology has details sufficient to allow experimental reproducibility. The reviewer has only a couple concerns as listed below.

1. When the authors describe or discuss inhibition in the olfactory bulb, only GCs and EPL INs were considered, totally ignoring the inhibitory glomerular neurons. Anatomical evidence shows comparable or higher number of INs in the glomerular layer than that of GCs (Parrish-Aungst et al, 2007). The functional impact of glomerular INs on OB output was also consistently reported, especially in the past decade or so. The glomerular short axon cells are also supposed to contribute to lateral inhibition but was not mentioned in this manuscript. This treatment of the literature seems to be unfair and needs correction. 

2. While the authors tried to emphasize perisomatic inhibition involving somata, proximal apical dendrites and axon hillocks, synaptic interactions between EPL FSIs and lateral dendrites of MTCs cannot be completely excluded, especially for lateral inhibition. Thus, this may also need to be discussed in the text.

3. Minor: cited references need to be verified. At least the following three look incomplete:

Line 1208: Marín O (2012) Interneuron dysfunction in psychiatric disorders. 13:107-120.

Line 1209-1210: Markram H, Toledo-Rodriguez M, Wang Y, Gupta A, Silberberg G, Wu C (2004) Interneurons of the neocortical inhibitory system. 5:793-807. 

Line 1271: Silver RA (2010) Neuronal arithmetic. 11:474-489.

Reviewer #3: Effective decision making necessitates efficient processing of sensory information at behaviorally relevant time scales. Numerous studies in the rodent olfactory system over the years have conclusively shown the time scale for such processing to be on the order of 100-200 ms. In this period, recurrent circuit computations with the olfactory bulb (OB) are thought to play an important role. However, paradoxically, most of the electrophysiological studies to date have provided evidence for recurrent inhibition to be very delayed, very long lasting and low fidelity. This current work provides evidence for a form of inhibition (short latency, low jitter and high probability) that could play an essential role in bridging the gap between these two phenomena. The discovery and characterization of a population of interneurons that mediates rapid, reliable and significant lateral (and auto) inhibition of principal cells in the OB is highly significant. The experiments are extensive and extremely well done. The discovery is likely to be foundational for olfactory neuroscience. There is very little to quibble. Below are some specific comments that might be helpful.

Introduction, Line 40: To be fair here, the authors may want to acknowledge all the hard work on glomerular layer interneurons by people like Shipley.

Line 76: The phrasing feels clunky - just say "wild-type mice" or such?

Line 79: While the general idea of 2 big classes seems reasonable, in reality, I suspect that the number of classes is a bit dependent on hyperparameters used in any classification algorithm.

Line 88 and other places further down: Instead of "Magnification", use some phrase such as "Magnified region"

Line 135: Are the authors able to offer any information on differences in the physical location of FSIs connected to TCs vs MCs?

Similarly, the authors show that the connection probability and strength did not differ between FSIs and MCs vs. tufted cells, did they observe any fast lateral inhibition between the two populations?

There are many instances of "Error! Reference source not found" - please fix

In the Discussion section, I wonder if the authors could discuss or speculate on what makes the fast (FSI mediated) and slow lateral inhibition to be mutually exclusive. Were there any specific features (~ distance, location, connected to same glomerulus or different glomeruli) that determine the type of lateral inhibition between MTC pairs

---

## [Editor Report · Decision Letter 2]

17 Jul 2024

Dear Dr Burton,

Thank you for your patience while we considered your revised manuscript "Fast-spiking interneuron detonation drives high-fidelity inhibition in the olfactory bulb" for publication as a Research Article at PLOS Biology. This revised version of your manuscript has been evaluated by the PLOS Biology editors and the Academic Editor.

Based on our Academic Editor's assessment of your revision, we are likely to accept this manuscript for publication, provided you satisfactorily address the following data and other policy-related requests: 

* Please move the S1 Figure to the main part, it could even be the first main figure as the current Figure 1 is already very dense. Reviewer 1 makes a valid case about accessibility and we think that having the clear and easily understandable scheme will clarify the context.

* DATA POLICY:

Regardless of the method selected, please ensure that you provide the individual numerical values that underlie the summary data displayed in the following figure panels as they are essential for readers to assess your analysis and to reproduce it: 1NSTU, 2C, 3G, 5G, 6G and similar panels in the supplementary information.

* CODE POLICY

* Please note that per journal policy, we do not allow the mention of "data not shown", "personal communication", "manuscript in preparation" or other references to data that is not publicly available or contained within this manuscript. Please either remove mention of these data or provide figures presenting the results and the data underlying the figure(s).

We expect to receive your revised manuscript within two weeks. 

*Published Peer Review History*

*Press*

Sincerely,

Christian

Christian Schnell, PhD

Senior Editor

cschnell@plos.org

PLOS Biology

---

## [Editor Report · Decision Letter 3]

26 Jul 2024

Dear Dr Burton,

Thank you for the submission of your revised Research Article "Fast-spiking interneuron detonation drives high-fidelity inhibition in the olfactory bulb" for publication in PLOS Biology. On behalf of my colleagues and the Academic Editor, Izumi Fukunaga, I am pleased to say that we can in principle accept your manuscript for publication, provided you address any remaining formatting and reporting issues. These will be detailed in an email you should receive within 2-3 business days from our colleagues in the journal operations team; no action is required from you until then. Please note that we will not be able to formally accept your manuscript and schedule it for publication until you have completed any requested changes.

PRESS

Sincerely, 

Christian

Christian Schnell, PhD

Senior Editor

PLOS Biology

cschnell@plos.org